# Logarithmic Regret for Online KL-Regularized Reinforcement Learning

**Heyang Zhao** [* 1]  **Chenlu Ye** [* 2]  **Wei Xiong** [2]  **Quanquan Gu** [1]  **Tong Zhang** [2]

## Abstract

Recent advances in Reinforcement Learning from Human Feedback (RLHF) have shown that KL-regularization plays a pivotal role in improving the efficiency of RL fine-tuning for large language models (LLMs). Despite its empirical advantage, the theoretical difference between KL-regularized RL and standard RL remains largely under-explored. While there is a recent line of work on the theoretical analysis of KL-regularized objective in decision making (Xiong et al., 2024a; Xie et al., 2024; Zhao et al., 2024), these analyses either reduce to the traditional RL setting or rely on strong coverage assumptions. In this paper, we propose an optimism-based KL-regularized online contextual bandit algorithm, and provide a novel analysis of its regret. By carefully leveraging the benign optimization landscape induced by the KL-regularization and the optimistic reward estimation, our algorithm achieves an $\mathcal{O}\big(\eta \log(N_{\mathcal{R}}T) \cdot d_{\mathcal{R}}\big)$ logarithmic regret bound, where $\eta, N_{\mathcal{R}}, T, d_{\mathcal{R}}$ denote the KL-regularization parameter, the cardinality of the reward function class, number of rounds, and the complexity of the reward function class. Furthermore, we extend our algorithm and analysis to reinforcement learning by developing a novel decomposition over transition steps and also obtain a similar logarithmic regret bound.

## 1. Introduction

We study the KL-regularized contextual bandit (Langford & Zhang, 2007; Xiong et al., 2024a) and Markov decision processes (MDPs) (Sutton, 2018) in this paper. These two frameworks have received significant attention due to their tremendous successes in the post-training stage of modern large language models (LLMs), which is commonly referred to as the *Reinforcement Learning from Human Feedback* (RLHF) (Bai et al., 2022; Ouyang et al., 2022). The RLHF learning paradigm has been used to make a powerful chatbot by aligning the LLMs with human preference, making the model generation helpful, honest, and harmless (Bai et al., 2022). Notable examples include the OpenAI's Chat-GPT (OpenAI, 2023), Anthropic's Claude (Bai et al., 2022), and Google's Gemini (Team et al., 2023). More recently, the KL-regularized RL framework has also been applied to enhance multi-turn reasoning capabilities, resulting in powerful reasoning models such as GPT4-o1 and DeepSeek-R1.

However, the RLHF process also faces challenges. It often causes a decline in certain abilities acquired during pre-training and supervised fine-tuning (SFT), a phenomenon commonly known as "alignment tax" (Askell et al., 2021; Lin et al., 2023). For instance, contrastive learning without regularization, as observed in Meng et al. (2024), can degrade performance on standard reasoning benchmarks like MATH (Hendrycks et al., 2021) and GSM8K (Cobbe et al., 2021). Additionally, unregularized RL training presents significant challenges on computational efficiency and training stability (Casper et al., 2023). To address these issues, practitioners often optimize a KL-regularized objective (defined formally in Section 4) to balance reward optimization and mitigate alignment tax.

Moreover, these KL-regularized frameworks often demonstrate superior sample efficiency compared to standard deep RL tasks. For example, models with 52B parameters (Claude (Bai et al., 2022)) and 671B parameters (DeepSeek-R1 (DeepSeek-AI et al., 2025)) achieve substantial policy improvement with only tens of thousands of samples or thousands of training steps. Similar observations apply to direct preference alignment algorithms (Rafailov et al., 2023; Tunstall et al., 2023; Chen et al., 2024; Dong et al., 2024), which develop state-of-the-art open-source chatbots using fewer than 100K samples. These results highlight the superior sample efficiency of KL-regularized RL, surpassing traditional deep RL applications (Silver et al., 2016) and their analyses in the RL theory literature.

However, despite the empirical success, a fundamental connection between KL regularization and learning efficiency has not been established so far theoretically. While some

---

[*]Equal contribution [1]University of California, Los Angeles, CA 90095, USA [2]University of Illinois Urbana-Champaign, IL 61801, USA. Correspondence to: Quanquan Gu <qgu@cs.ucla.edu>, Tong Zhang <tongzhang@tongzhang-ml.org>.

*Proceedings of the $42^{nd}$ International Conference on Machine Learning*, Vancouver, Canada. PMLR 267, 2025. Copyright 2025 by the author(s).

recent studies have attempted to analyze these frameworks, they often rely on standard analysis techniques, yielding regret guarantees similar to those of standard RL (Xiong et al., 2024a; Xie et al., 2024; Xiong et al., 2024b; Cen et al., 2024), or they depend on strong coverage assumptions and restricted to the bandit settings (Zhao et al., 2024). A notable exception is the approach of Tiapkin et al. (2023; 2024), which achieves $\widetilde{\mathcal{O}}(H^5 d^2 \eta/\epsilon)$ sample complexity for KL-regularized linear MDPs, and $\widetilde{\mathcal{O}}(H^5 S^2 A \eta/\epsilon)$ sample complexity for tabular MDPs. However, their analysis focuses on the *pure exploration* or *best policy identification* setting, where the goal is to find near-optimal policies using the least possible amount of interactions with the environment, rather than the online setting where the agent needs to trade off between exploration and exploitation. Therefore, the following pivotal question remains open:

*Is KL-regularized RL more efficient than standard RL in the online setting without additional coverage assumption?*

In this work, we address this question by designing provably efficient algorithms based on the optimism in the face of uncertainty (OFU) principle and develop **refined policy suboptimimality decomposition** for both contextual bandits and MDPs.

We establish the theoretical guarantees for the algorithms and demonstrate their statistical advantages over standard RL scenarios. We summarize our contributions as follows.

- For KL-regularized contextual bandits, we establish the first $\mathcal{O}\big(\eta \log(N_{\mathcal{R}} T) \cdot d_{\mathcal{R}}\big)$ regret bound that scales logarithmically with time steps $T$ in the standard online RL setting, where $\eta$ is the KL-regularization parameter, $N_{\mathcal{R}}, d_{\mathcal{R}}$ denote the cardinality of the reward function class $\mathcal{R}$, and its eluder dimension. This result significantly improves upon the previous $\mathcal{O}(\sqrt{T})$ bound (Xiong et al., 2024a) and eliminates the strong coverage condition in prior work (Zhao et al., 2024).

- Distinct from the previous analyses that solely rely on the learned policy maximizing the KL-regularized objective, we take a novel approach by expressing the suboptimality gap in terms of a functional gap with respect to the policy $\pi_R$ induced by a proxy reward function $R$. With a fine-grained analysis for the derivative of the gap, we then establish the monotonicity in the sub-optimality gap via the optimistic reward estimation. This allows us to obtain the uncertainty induced by the policy at the current time step so that the sum of squared uncertainty can be bounded by the eluder dimension.

- We extend these techniques to KL-regularized MDPs and establish the first $\mathcal{O}(\log T)$ regret bound in the literature. The key to this improved regret bound is a novel policy decomposition technique through multi-steps. These techniques may be of independent interest and have the potential to inspire future research on KL-regularized decision-making problems.

## 2. Related Work

**RLHF.** Reinforcement Learning from Human Feedback (RLHF) has achieved tremendous successes in the modern large language model post training (OpenAI, 2023; Bai et al., 2022; Ouyang et al., 2022; Team et al., 2023). The dominant approach in the area is based on the reward training and policy optimization with the PPO algorithm (Schulman et al., 2017). However, applying PPO effectively in the context of LLMs presents significant challenges (Choshen et al., 2019). However, getting the PPO work is challenging in the context of LLMs (Choshen et al., 2019). In view of this, researchers have spent great efforts in proposing alternative approaches to the PPO algorithm.

One line of research revisits REINFORCE-based variants such as ReMAX and GRPO (Li et al., 2023a; Shao et al., 2024), with the KL-regularized objective. Another approach focuses on direct preference learning (Zhao et al., 2023b; Rafailov et al., 2023; Azar et al., 2023), which bypasses the reward modeling stage and directly optimizes the policy using the preference dataset in a supervised manner. A notable example is the Direct Preference Optimization (DPO) algorithm (Rafailov et al., 2023), which has gained great attention in both the open-source community (Tunstall et al., 2023; Lambert et al., 2024) and industrial applications such as Llama (Dubey et al., 2024). All approaches mentioned above are derived under the KL-regularized framework studied in this paper. An exception to this trend is best-of-n (BoN) sampling and rejection sampling fine-tuning (Bai et al., 2022; Dong et al., 2023; Touvron et al., 2023), where a reward model is used to filter samples for final output or select training samples. However, recent works show that the success of BoN sampling may essentially result from the fact that it is optimal in terms of the KL-reward trade-off (Gui et al., 2024; Yang et al., 2024).

**Theory of RLHF.** The theoretical foundation of RLHF traces back to dueling bandits (e.g., Yue et al., 2012; Saha, 2021; Bengs et al., 2021), which studied preference feedback in non-regularized settings. This was later extended to online reinforcement learning with finite state spaces

---

[1] Xiong et al. (2024a) studies relative-preference feedback, so their sample complexity additionally depends on $e^\eta$. Since our work focuses on absolute rewards, we omit this dependence.

[2] Xiong et al. (2024b) considers the trajectory-level reward and learns the reward in $\mathcal{R}$ and transition probability in $\mathcal{P}$ separately, so the dependence on $d_{\mathcal{R}}, N_{\mathcal{R}}$ and $d_c P, N_{\mathcal{P}}$ are separete.

[3] Xie et al. (2024) considers the deterministic-transition setting and optimizes the policy directly.

*Table 1.* Comparison of online KL-regularized algorithms in bandits and MDPs, where $\epsilon > 0$ is the sub-optimality gap, $T$ is the number of rounds, $d$ is the vector dimension for linear models, constant $\eta > 0$ is the KL-regularized parameter, $C_{\mathrm{GL}}$ is the global coverage condition, $d_{\mathcal{R}}$ and $d_{\mathcal{F}}$ are the complexity measure for general function class $\mathcal{R}$ and $\mathcal{F}$, $\widetilde{\mathcal{O}}$ omits the logarithmic order for $T$ and $1/\epsilon$. The notations $N, N_{\mathcal{R}}, N_{\mathcal{F} \oplus \mathcal{B}}, N_{\Pi}$ represent the cardinality or covering number for the reward, value and bonus function classes. The fourth column represents the number of samples needed to achieve an $\epsilon$-suboplimality, and the last column shows whether they require the coverage condition. We convert the regret bounds of our algorithms to sample complexity by Lemma D.2. We remark that the results of Xiong et al. (2024a); Xie et al. (2024); Xiong et al. (2024b) are based on preference feedback. When comparing with them, we mainly focus on the *order* of dependency on $T$. See Remark 3.1 for further discussion on the extension to preference feedback.

| Setting | Algorithm | Regret | Sample Complexity | Coverage |
|---|---|---|---|---|
| **Bandits** | Online Iterative GSHF[1] (Xiong et al., 2024a) | – | $\widetilde{\mathcal{O}}\big(d^2/\epsilon^2\big)$ | × |
| | Two-Stage Mixed-Policy Sampling (Zhao et al., 2024) | – | $\widetilde{\mathcal{O}}\big((\eta^2 C_{\mathrm{GL}}^2 + \eta/\epsilon)\log N_{\mathcal{R}}\big)$ | ✓ |
| | KL-UCB (Theorem 4.1) | $\mathcal{O}\big(\eta d_{\mathcal{R}}\log T \cdot \log(N_{\mathcal{R}})\big)$ | $\widetilde{\mathcal{O}}\big(\eta d_{\mathcal{R}} \cdot \log(N_{\mathcal{R}})/\epsilon\big)$ | × |
| **MDPs** | Online Iterative M-GSHF[2] (Xiong et al., 2024b) | $\widetilde{\mathcal{O}}\big(\sqrt{d_{\mathcal{R}}T \log N_{\mathcal{R}}} + H d_{\mathcal{P}}\log N_{\mathcal{P}}\big)$ | $\widetilde{\mathcal{O}}\big(d_{\mathcal{R}}\log N_{\mathcal{R}}/\epsilon^2 + H d_{\mathcal{P}}\log N_{\mathcal{P}}/\epsilon\big)$ | × |
| | XPO (Xie et al., 2024)[3] | – | $\widetilde{\mathcal{O}}\big(d_{\mathcal{F}}\log N_{\mathcal{F}}/\epsilon^2\big)$ | × |
| | KL-LSVI-UCB (Theorem 5.1) | $\mathcal{O}\big(\eta H^2 d_{\mathcal{F}}\log T \cdot \log N_{\mathcal{F} \oplus \mathcal{B}}\big)$ | $\widetilde{\mathcal{O}}\big(\eta H^2 d_{\mathcal{F}} \cdot \log N_{\mathcal{F} \oplus \mathcal{B}}/\epsilon\big)$ | × |

(tabular settings) and function approximation (Xu et al., 2020; Novoseller et al., 2020; Pacchiano et al., 2021; Chen et al., 2022). More recently, Zhan et al. (2023b); Wu & Sun (2023) developed reward-free learning algorithms and sampling-based methods for online RLHF. For the offline learning, Zhu et al. (2023); Zhan et al. (2023a); Li et al. (2023b); Zhong et al. (2024); Huang et al. (2024) propose sample-efficient algorithms under suitable coverage conditions. These works mainly develop techniques to estimate the underlying reward model associated with the Bradley-Terry model from querying the preference oracle (human) and achieve similar order regret with the standard reward learning. However, since they only consider reward maximization, the results deviate from the practical applications of RLHF. For example, reward maximization frameworks often assume a deterministic optimal policy, which is unsuitable for generative models.

After these works, the recent project Xiong et al. (2024a) provides the first provably efficient algorithm of RLHF under the KL-regularized contextual bandit formulation. The result is further refined in (Xie et al., 2024) and Xie et al. (2024); Liu et al. (2024); Cen et al. (2024) propose provably efficient algorithms with optimistically biased optimization targets, which originate from the feel-good sampling (Zhang, 2022). In parallel, Wang et al. (2023); Ye et al. (2024) extend the techniques of preference learning to the general preference setting under the Markov game formulation. However, their techniques simply discard the KL divergence in the target, and use the standard techniques to get results that are similar to the non-regularized problems, which are essentially sub-optimal for the KL-regularized framework. In contrast, in this work, we aim to leverage the structure of

the KL-regularized problem and develop new techniques and algorithms that achieve superior theoretical guarantees compared to prior studies.

The most closely related work to our project is Zhao et al. (2024), which considers the KL-regularized RL in the bandit setting. They propose a two-stage mixed-policy sampling algorithm and provide a regret bound which enjoys an $\mathcal{O}(1/\epsilon)$ sample complexity. However, their results rely on a relatively strong coverage assumption, which is not compatible with the practical applications of RLHF. In contrast, our work provides a novel algorithm and analysis that achieves a logarithmic regret bound without the coverage assumption. We summarize the comparison of our work with the existing literature in Table 1.

**Notation.** For a finite function class $\mathcal{F}$, we use $N_{\mathcal{F}}$ to represent its cardinality. We use $\widetilde{\mathcal{O}}$ to omit the logarithmic orders. We use the convention $[n] = \{1, \ldots, n\}$. For any vector $x$ and a matrix $\Sigma$, let $\|x\|_{\Sigma} = \sqrt{x^\top \Sigma x}$. For a function $R : \mathcal{X} \times \mathcal{A} \to \mathbb{R}$, parameter $\eta > 0$ and the reference policy $\pi_{\mathrm{ref}}$, let the normalization constant $Z_R(x) = \mathbb{E}_{a \sim \pi_{\mathrm{ref}}(\cdot|x)} \exp(\eta R(x, a))$.

## 3. Background

### 3.1. KL-Regularized Contextual Bandits

The contextual bandit problem with KL-regularized objective is defined as follows.

At each round $t \geq 1$, the learner observes a context $x_t \in \mathcal{X}$, and chooses an action $a_t \in \mathcal{A}$, where $\mathcal{X}$ is the context space and $\mathcal{A}$ is the action space. The learner then receives a

reward $r_t \in \mathbb{R}$ which characterizes the gain of the learner from choosing action $a_t$ under context $x_t$. We assume that the reward $r_t$ is generated by

$$r_t = R^*(x_t, a_t) + \epsilon_t,$$

where $R^*(x_t, a_t)$ is the unknown reward function and $\epsilon_t$ is an independent 1-sub-Gaussian zero-mean random noise. In practice, the context can be a prompt, and the action is the response generated by the LLMs. The reward signal is a sentence reward that is commonly used (Ouyang et al., 2022; Bai et al., 2022; Touvron et al., 2023).

*Remark* 3.1. We do not consider the preference feedback here since the more recent applications in building reasoning models (DeepSeek-AI et al., 2025) and improving LLM safety (Guan et al., 2024) assume the existence of the absolute reward value (e.g. the binary reward indicator of the correctness). The results presented in this work can be readily extended to the preference feedback using the techniques developed in dueling bandit or RL (Yue et al., 2012; Pacchiano et al., 2021; Xiong et al., 2024a; Li et al., 2024).

**Assumption 3.2** (Reward function realizability). Assume that there exists a reward function class $\mathcal{R} : \mathcal{X} \times \mathcal{A} \to [0, 1]$ such that $R^*(x, a) \in \mathcal{R}$.

Without loss of generality, we assume that the function class has finite cardinality $|\mathcal{R}|$, and we can generalize the analysis to an infinite function class by considering a covering number (Zhang, 2023).

**Definition 3.3** (Uncertainty and eluder dimension). For any sequence $\mathcal{D}_{t-1} = \{(x_i, a_i)\}_{i=1}^{t-1}$, we define the uncertainty of $(x, a)$ with respect to $\mathcal{R}$ as:

$$U_{\mathcal{R}}(\lambda, x, a; \mathcal{D}_{t-1})$$
$$= \sup_{R_1, R_2 \in \mathcal{R}} \frac{|R_1(x, a) - R_2(x, a)|}{\sqrt{\lambda + \sum_{i=1}^{t-1} \left( R_1(x_i, a_i) - R_2(x_i, a_i) \right)^2}}.$$

Then, the eluder dimension is given by $d(\mathcal{R}, \lambda, T) := \sup_{x_{1:T}, a_{1:T}} \sum_{t=1}^{T} \min \left( 1, [U_{\mathcal{R}}(\lambda, x_t, a_t; \mathcal{D}_{t-1})]^2 \right)$.

The uncertainty $U_{\mathcal{R}}(\lambda, x, a; \mathcal{D}_t)$ measures how much difference the new-coming data $x, a$ has with the history data $\mathcal{D}_t$, and is widely adopted in RL literature with general function approximation (Zhang, 2023; Ye et al., 2023; Agarwal et al., 2023; Zhao et al., 2023a). To illustrate it better, we use the linear function as a special case, where the function class $\mathcal{R}$ can be embedded into a linear mapping $\mathcal{R} = \{\theta^\top \phi(\cdot, \cdot) : \theta \in \mathbb{R}^d, \|\theta\|_2 \leq B\}$. Let the covariance matrix $\Sigma_t = \sum_{i=1}^{t} \phi(x_i, a_i)\phi(x_i, a_i)^\top + \lambda/B \cdot I$. Then,

the eluder coefficient can be simplified as

$$U_{\mathcal{R}}(\lambda, x, a; \mathcal{D}_t)$$
$$= \sup_{\theta_1, \theta_2 \in \mathbb{R}^d} \frac{|((\theta_1 - \theta_2)^\top \phi(x, a))|}{\sqrt{\lambda + \sum_{i \in [t]} ((\theta_1 - \theta_2)^\top \phi(x_i, a_i))^2}}$$
$$\leq \sup_{\theta_1, \theta_2 \in \mathbb{R}^d} \frac{|((\theta_1 - \theta_2)^\top \phi(x, a))|}{\sqrt{(\theta_1 - \theta_2)^\top \Sigma_t (\theta_1 - \theta_2)}} \leq \left\| \phi(x, a) \right\|_{\Sigma_t^{-1}},$$

where the inequality uses the Cauchy–Schwarz inequality. Hence, the uncertainty reflects how much a direction in the feature space has been explored. Further, the eluder dimension represents how many times the model can be "surprised" by the out-of-sample data over $T$ rounds. One can show that this definition is more general than the original one in (Russo & Van Roy, 2013) as we further take the magnitude of violation into consideration while the original eluder dimension only counts the frequency (Zhang, 2023; Xie et al., 2022).

We consider a KL-regularized **objective** as follows:

$$J(\pi) := \mathbb{E}_{x \sim d_0} \mathbb{E}_{a \sim \pi(\cdot|x)} \left[ R^*(x, a) - \frac{1}{\eta} \log \frac{\pi(a|x)}{\pi_{\text{ref}}(a|x)} \right]$$
$$= \mathbb{E}_{x \sim d_0} \mathbb{E}_{a \sim \pi(\cdot|x)} [R^*(x, a)] - \frac{1}{\eta} \text{KL}\left( \pi(\cdot|x) \| \pi_{\text{ref}}(\cdot|x) \right),$$

where $\pi_{\text{ref}}$ is the reference policy known to the learner, and $\eta > 0$ is the regularization parameter. This formulation is adopted in nearly all RLHF practice for LLM alignment (Bai et al., 2022; Ouyang et al., 2022; Touvron et al., 2023), and there is a closed-form solution for this optimization, also known as Gibbs distribution.

**Lemma 3.4** (Solution of KL-regularized Optimization (Proposition 7.16 and Theorem 15.3 of Zhang (2023)). *For any fixed $x \in \mathcal{X}$ and reward function $R$, we have*

$$\max_{\pi} \mathbb{E}_{a \sim \pi(\cdot|x)} \left[ R(x, a) - \eta^{-1} \text{KL}\left( \pi(\cdot|x) \| \pi_{\text{ref}}(\cdot|x) \right) \right]$$
$$= \frac{1}{\eta} \cdot \log \mathbb{E}_{a \sim \pi_{\text{ref}}(\cdot|x)} \exp \left( \eta R(x, a) \right),$$

*where $Z_R(x)$ is the normalization constant and the minimizer of the loss functional is*

$$\pi_R^\eta(a|x) = \frac{1}{Z_R(x)} \pi_{\text{ref}}(a|x) \exp \left( \eta R(x, a) \right).$$

### 3.2. KL-Regularized Reinforcement Learning

In this section, we introduce the KL-regularized MDP problem. A Markov Decision Process (MDP) is defined by a tuple $\mathcal{M} = (\mathcal{S}, \mathcal{A}, H, \mathbb{P}, d_0, r)$, where $\mathcal{S}$ is the state space, $\mathcal{A}$ is the action space, $H$ is the time horizon, transition probability $\mathbb{P} = \{\mathbb{P}_h\}_{h=1}^H$ denotes the probability $\mathbb{P}(s_{h+1}|s_h, a_h)$ of transition from the current $(s_h, a_h)$ to the next state $s_{h+1}$ at each step $h$, $d_0$ is the initial state distribution, and $r = \{r_h : \mathcal{S} \times \mathcal{A} \to \mathbb{R}\}_{h=1}^H$ is the reward function.

A policy $\pi$ is a sequence of function $\pi = \{\pi_1, \ldots, \pi_h\}$ with $\pi_h : \mathcal{S} \times \mathcal{A} \to [0,1]$ for all $h \in [H]$. The **objective** is to find a policy $\pi$ that maximizes the following KL-regularized value:

$$J(\pi) = \mathbb{E}^\pi \left[ \sum_{h=1}^{H} r_h(s_h, a_h) - \eta^{-1} \sum_{h=1}^{H} \log \frac{\pi_h(a_h|s_h)}{\pi_{\text{ref},h}(a_h|s_h)} \right],$$

where $\mathbb{E}^\pi$ denotes the expectation $\{(s_h, a_h, r_h)\}_{h=1}^{H}$ of the reward and the trajectory induced by policy $\pi$ from the initial state $s_1 \sim d_0$.

We can define the value functions as the expected future return with KL-regularization:

$$V_h^\pi(s_h) = \sum_{h'=h}^{H} \mathbb{E}^\pi \Big[ r_{h'}(s_{h'}, a_{h'})$$
$$- \frac{1}{\eta} \cdot \text{KL}\big(\pi_{h'}(\cdot|s_{h'}) \| \pi_{\text{ref},h'}(\cdot|s_{h'})\big) \,\Big|\, s_h \Big],$$

$$Q_h^\pi(s_h, a_h) = r_h(s_h, a_h) + \mathbb{E}^\pi \sum_{h'=h+1}^{H} \Big[ r_{h'}(s_{h'}, a_{h'})$$
$$- \frac{1}{\eta} \cdot \text{KL}\big(\pi_{h'}(\cdot|s_{h'}) \| \pi_{\text{ref},h'}(\cdot|s_{h'})\big) \,\Big|\, s_h, a_h \Big].$$

We can also iteratively define the regularized value function as follows: $V_{H+1}^\pi(s_{H+1}) = 0$, and

$$V_h^\pi(s_h) = \mathbb{E}_{a_h \sim \pi_h(\cdot|s_h)} \Big[ Q_h^\pi(s_h, a_h)$$
$$- \frac{1}{\eta} \cdot \text{KL}\big(\pi_h(\cdot|s_h) \| \pi_{\text{ref},h}(\cdot|s_h)\big) \Big],$$

$$Q_h^\pi(s_h, a_h) = r_h(s_h, a_h) + \mathbb{E}_{s_{h+1} \sim \mathbb{P}_h(\cdot|s_h, a_h)} \Big[ V_{h+1}^\pi(s_{h+1}) \Big].$$

We further define the optimal value function $\{V_h^*\}_{h \in [H]}$ and the optimal action-value function $\{Q_h^*\}_{h \in [H]}$ as

$$V_h^*(s_h) = \max_\pi V_h^\pi(s_h), \quad Q_h^*(s_h, a_h) = \max_\pi Q_h^\pi(s_h, a_h).$$

Assume optimal policy is achieved at $\pi^*$, using Lemma 3.4 and a backward iteration starting from the $V_{H+1}^*(s_{H+1}) = 0$, we have the following proposition.

**Proposition 3.5.** *The optimal policy is a layer-wise Gibbs distribution of the $Q^*$:*

$$\pi_h^*(a_h|s_h) = \frac{1}{Z_h^*(s_h)} \pi_{\text{ref},h}(a_h|s_h) \cdot \exp\big(\eta Q_h^*(s_h, a_h)\big),$$

*where $Z_h^*(s_h) := \mathbb{E}_{a_h \sim \pi_{\text{ref},h}(\cdot|s_h)} \exp\big(\eta Q_h^*(s_h, a_h)\big)$ is the normalization constant. Also, the V value is the maximum value of the single-step KL-regularized optimization problem:*

$$V_h^*(s_h) = \frac{1}{\eta} \log \mathbb{E}_{a_h \sim \pi_{\text{ref},h}(\cdot|s_h)} \exp\big(\eta Q_h^*(s_h, a_h)\big). \quad (1)$$

*We also have the following connection between the $Q^*$ and $V^*$:*

$$Q_h^*(s_h, a_h) = r_h(s_h, a_h) + \mathbb{E}_{s_{h+1} \sim \mathbb{P}_h(\cdot|s_h, a_h)} V_{h+1}^*(s_{h+1}). \quad (2)$$

Without loss of generality, we assume that $Q_h^*(s_h, a_h) \in [0,1]$ for any $(s_h, a_h) \in \mathcal{S} \times \mathcal{A}$[1]. From (1), we know that $V_h^*(s_h) \in [0,1]$ for any $s_h \in \mathcal{A}$.

**Function approximation** We approximate the value function $\{Q_h^*\}_{h \in [H]}$ by a function class $\mathcal{F} = \mathcal{F}_1 \times \ldots \times \mathcal{F}_H$ where $\mathcal{F}_h : \mathcal{S} \times \mathcal{A} \to [0,1]$. Similarly, we can define related optimal policies and value functions with the estimated $Q$-function $\widehat{Q}_h \in \mathcal{F}_h$ and $V$-function $\widehat{V}_h$.

Then, we define the KL-regularized Bellman operator $\mathcal{T}_\eta^h$ on space $\mathcal{F}$: for any $f_{h+1} \in \mathcal{F}_{h+1}$,

$$\mathcal{T}_\eta^h f_{h+1}(s_h, a_h) := r_h(s_h, a_h) + \frac{1}{\eta} \mathbb{E}_{s_{h+1}|s_h, a_h} V_{h+1}^f(s_{h+1}),$$

where $V_{h+1}^f(s_{h+1}) = \eta^{-1} \log \mathbb{E}_{a_{h+1} \sim \pi_{\text{ref},h+1}(\cdot|s_{h+1})} \exp\big(\eta f_{h+1}(s_{h+1}, a_{h+1})\big)$.

To ensure efficient learning, we suppose that the true $Q$-value function belongs to the considered function class and the one-step backward of the value function also remains in the considered function class, which are known as realizability and Bellman completeness assumptions, respectively.

**Assumption 3.6** (Realizability). For each $h \in [H]$, we have $Q_h^* \in \mathcal{F}_h$.

**Assumption 3.7** (Bellman completeness). For each $h \in [H]$ and any $f_{h+1} \in \mathcal{F}_{h+1}$, we have $\mathcal{T}_\eta^h f_{h+1} \in \mathcal{F}_h$.

These two assumptions are standard and normal in RL literature with general function approximation (Wang et al., 2020; Jin et al., 2021; Zhang, 2023).

# 4. KL-Regularized Contextual Bandits

## 4.1. The Proposed Algorithm and Result

We develop the KL-regularized Upper Confidence Bound (KL-UCB) algorithm in Algorithm 1. In each round $t \in [T]$, after observing context $x_t \sim d_0$, taking action $a_t \sim \pi_t(\cdot|x_t)$, and receiving the reward $r_t = R^*(x_t, a_t) + \epsilon_t$, we compute the estimator $\widehat{R}_t$ by solving the least square regression. Then, we perform optimism by adding the following bonus term to the reward estimator:

$$b_t(x, a) = \min\left\{1, \beta_T \cdot U_{\mathcal{R}_t}(\lambda, x, a; \mathcal{D}_t)\right\}, \quad (3)$$

where $U_{\mathcal{R}_{t-1}}$ is the uncertainty in Definition 3.3, $\mathcal{D}_{t-1} = \{(x_i, a_i)\}_{i \in [t-1]}$, and we define the confidence set iteratively:

$$\mathcal{R}_t = \{R \in \mathcal{R} : \sum_{i=1}^{t} (R(x_i, a_i) - \widehat{R}_t(x_i, a_i))^2 + \lambda \leq \beta_T^2\}.$$

---

[1]This is for the simplicity of analysis. By multiplying the results by $H$, our analysis can be applied to the cases where $r_h \in [0,1]$ for $h \in [H]$.

**Algorithm 1** KL-Regularized UCB

1: **Input:** $\eta$, $\epsilon$, $\pi_{\text{ref}}$, $\mathcal{R}$.
2: **for** $t = 1, \ldots, T$ **do**
3:  Sample context $x_t \sim d_0$ and action $a_t \sim \pi_t(\cdot|x_t)$.
4:  Observe reward $r_t = R^*(x_t, a_t) + \epsilon_t$, where $\epsilon_t$ is the random noise.
5:  Compute the least square estimate of the reward function based on $\mathcal{D}_t = \{(x_i, a_i, r_i)\}_{i=1}^{t}$:

$$\widehat{R}_t \leftarrow \operatorname*{argmin}_{R \in \mathcal{R}} \sum_{i=1}^{t} (R(x_i, a_i) - r_i)^2.$$

6:  Apply the planning oracle to compute $\pi_{t+1}(\cdot|\cdot) \propto \pi_{\text{ref}}(\cdot|\cdot) \exp\left[\eta\left(\widehat{R}_t(\cdot, \cdot) + b_t(\cdot, \cdot)\right)\right]$, where $b_t$ is the exploration bonus term defined in (3).
7: **end for**

**Theorem 4.1.** *Under Assumption 3.2, for any $\delta > 0$, by taking $\beta_T = 16 \log(N_{\mathcal{R}} T/\delta)$, with probability at least $1 - \delta$, the output of Algorithm 1 satisfies*

$$\text{Regret}(T) = \mathcal{O}\Big(\eta \log(N_{\mathcal{R}} T/\delta) \cdot d(\mathcal{R}, \lambda, T)\Big).$$

*Remark* 4.2. Theorem 4.1 establishes that the regret of Algorithm 1 scales logarithmically with $T$, rather than at the typical $\mathcal{O}(\sqrt{T})$ rate, thereby improving upon the previous analysis in Xiong et al. (2024a) under the standard online RL setting. To the best of our knowledge, this is the first analysis of a KL-regularized contextual bandit that achieves this logarithmic regret bound.

### 4.2. Proof Outline of Theorem 4.1

We first explain where the previous methods loosen and then highlight our methods to get the sharp bound.

**Review of Analysis in Previous Work** Previous analysis (Xiong et al., 2024a; Ye et al., 2024; Song et al., 2024) neglects the pivotal role of KL-regularization in the suboptimality decomposition and reduce to the traditional bandit analysis. Specifically, using the short-hand notation $R(x, \pi) = \mathbb{E}_{a \sim \pi(\cdot|x)} R(x, a)$ and $\text{KL}(\pi \| \pi') = \text{KL}(\pi(\cdot|x) \| \pi'(\cdot|x))$, they derive that

$$R^*(x, \pi^*) - \text{KL}(\pi^* \| \pi_{\text{ref}}) - \left(R^*(x, \pi_t) - \text{KL}(\pi_t \| \pi_{\text{ref}})\right)$$
$$= \underbrace{R(x, \pi^*) - \widehat{R}_t(x, \pi^*) - b_t(x, \pi^*)}_{\leq 0 \text{ (By Optimism)}}$$
$$+ \underbrace{\widehat{R}_t(x, \pi_t) - R^*(x, \pi_t) + b_t(x, \pi_t)}_{\leq 2b_t(x, \pi_t)}$$
$$+ \left(\widehat{R}_t(x, \pi^*) + b_t(x, \pi^*) - \text{KL}(\pi^*(\cdot|x) \| \pi_{\text{ref}}(\cdot|x))\right)$$
$$- \left(\widehat{R}_t(x, \pi_t) + b_t(x, \pi^*) - \text{KL}(\pi_t(\cdot|x) \| \pi_{\text{ref}}(\cdot|x))\right),$$

where on the right-hand side of the last equality, the first two terms follow from the optimistic reward estimator and

the definition of the bonus, the subtraction of the last two terms is upper bounded by $0$ since $\pi_t$ is the maximizer of the KL-regularized reward according to Algorithm 1. Essentially, deserting the last two terms is where they loosen and reduce to the traditional analysis. As a result, their regret is upper bounded by the first-order summation of bonuses $2 \sum_{t=1}^{T} \mathbb{E}_{x \sim d_0} b_t(x, \pi_t)$ and finally get an $\widetilde{\mathcal{O}}(\sqrt{T})$ regret.

**Our Analysis.** We provide the proof sketch of Theorem 4.1 here. The details are provided in Appendix A. Our analysis is divided into the following four parts.

**Part I: Single-Step Regret Decomposition.** For all $t \geq 1$, with $R^*(x, a) = \frac{1}{\eta} \log \exp\left(\eta R^*(x, a)\right)$ and the closed-form solution of $\pi^*$ and $\pi$ from Lemma 3.4, we have the following equality for the suboptimality gap:

$$
\begin{aligned}
&\mathbb{E}_{\pi^*}\left[R^*(x, a) - \frac{1}{\eta} \log \frac{\pi^*(a|x)}{\pi_{\text{ref}}(a|x)}\right] \\
&\quad - \mathbb{E}_{\pi_t}\left[R^*(x, a) - \frac{1}{\eta} \log \frac{\pi_t(a|x)}{\pi_{\text{ref}}(a|x)}\right] \\
&= \frac{1}{\eta} \log Z_{R^*}(x) - \frac{1}{\eta} \log Z_{\widehat{R}_{t-1} + b_{t-1}}(x) \\
&\quad + \mathbb{E}_{\pi_t}\left[\widehat{R}_{t-1}(x, a) - R^*(x, a) + b_{t-1}(x, a)\right],
\end{aligned}
\tag{4}
$$

where $Z_R(x)$ is the normalization term for the policy with respect to reward function $R$. We defer the detailed derivation of (4) to Appendix A.

*Remark* 4.3. Compared to previous work (Xiong et al., 2024a; Xie et al., 2024; Ye et al., 2024) that neglect the KL term and reduce to the traditional RL analysis, the upper bound in (4) is more refined and involves the gap of the normalization term $Z_R(x)$, which is crucial for the analysis of the KL-regularized objective.

**Part II: Shape of Regularized Suboptimality Gap.** To further analyze (4), we define a function

$$\Delta(x, R) := -\frac{1}{\eta} \log Z_R(x) + \mathbb{E}_{\pi_R^{\eta}}\left[R(x, a) - R^*(x, a)\right]. \tag{5}$$

and write (4) as the function gap:

$$\mathbb{E}_{x \sim d_0}[\Delta(x, \widehat{R}_{t-1} + b_{t-1}) - \Delta(x, R^*)]. \tag{6}$$

To analyze the behavior of (6), we have the following lemma, which reveals the gradient of $\Delta(x, R)$ with respect to $R$.

**Lemma 4.4.** *Let $Z_R(x) = \sum_{a \in \mathcal{A}} \pi_{\text{ref}}(a|x) \cdot \exp(\eta R(x, a))$ and $\pi_R^{\eta}(a|x) = \pi_{\text{ref}}(a|x) \exp(\eta R(x, a))/Z_R(x)$. We have the following partial gradient result:*

$$
\begin{aligned}
\frac{\partial \Delta(x, R)}{\partial R(x, a)} &= \eta \pi_R^{\eta}(a|x) \cdot \left(R(x, a) - R^*(x, a)\right) \\
&- \eta \sum_{a' \in \mathcal{A}} \pi_R^{\eta}(a|x) \cdot \pi_R^{\eta}(a'|x) \cdot \left(R(x, a') - R^*(x, a')\right).
\end{aligned}
\tag{7}
$$

The proof of this lemma is deferred to Appendix A. With the gradient information above, we can apply the mean value theorem to show that there exists an $f_\lambda = \lambda \cdot (\widehat{R}_{t-1} + b_{t-1}) + (1 - \lambda) \cdot R^*$ for some $\lambda \in (0, 1)$ such that

$$
\mathbb{E}_{x \sim d_0}[\Delta(x, \widehat{R}_{t-1} + b_{t-1}) - \Delta(x, R^*)]
$$
$$
= \mathbb{E}_{x \sim d_0} \Big[ \sum_{a \in \mathcal{A}} \Big( \eta \pi^\eta_{f_\lambda}(a|x) \cdot \big(f_\lambda(x, a) - R^*(x, a)\big)
$$
$$
- \eta \sum_{a' \in \mathcal{A}} \pi^\eta_{f_\lambda}(a|x) \cdot \pi^\eta_{f_\lambda}(a'|x) \cdot \big(f_\lambda(x, a') - R^*(x, a')\big) \Big)
$$
$$
\cdot \big(\widehat{R}_{t-1}(x, a) + b_{t-1}(x, a) - R^*(x, a)\big) \Big],
$$

**Part III: Benefits of Optimism.** The bonus used in Algorithm 1 ensures that with high probability, the true reward function is upper bounded by the optimistic reward function, which is formally stated in the following lemma.

**Lemma 4.5.** *Under Algorithm 1 and Assumption 3.2 and the condition that the noises $\epsilon_t$ are conditional 1-sub-Gaussian, we have with probability at least $1 - \delta$ for all $t \in [T]$, the uniform optimism event that $\mathcal{E}^t_{\mathrm{opt}} = \{\widehat{R}_t(x, a) + b_t(x, a) - R^*(x, a) \geq 0, \forall(x, a) \in \mathcal{X} \times \mathcal{A}\}$ holds true.*

Hence, conditioning on $\mathcal{E}^t_{\mathrm{opt}}$ we have $f_\lambda(x, a') - R^*(x, a') \geq 0, \quad \forall(x, a) \in \mathcal{X} \times \mathcal{A}$, which is substituted back into (8) to get

$$
\mathbb{E}_{x \sim d_0}[\Delta(x, \widehat{R}_{t-1} + b_{t-1}) - \Delta(x, R^*)]
$$
$$
\leq \eta \mathbb{E}_{x \sim d_0} \Big[ \sum_{a \in \mathcal{A}} \big(\widehat{R}_{t-1}(x, a) + b_{t-1}(x, a) - R^*(x, a)\big)
$$
$$
\cdot \underbrace{\big(f_\lambda(x, a) - R^*(x, a)\big) \cdot \pi^\eta_{f_\lambda}(a|x)}_{U(x,a)} \Big] \tag{8}
$$
$$
\leq \eta \mathbb{E}_{x \sim d_0} \mathbb{E}_{a \sim \pi^\eta_{\widehat{R}_{t-1} + b_{t-1}}(\cdot|x)} \Big[
$$
$$
\big(\widehat{R}_{t-1}(x, a) + b_{t-1}(x, a) - R^*(x, a)\big)^2 \Big],
$$

where the last inequality holds since by taking the derivative of $U(x, a)$ with respect to $\lambda$, we can show that $U(x, a)$ reaches the maximum when $\lambda = 1$.

**Part IV: Regret Bound.** Conditioning on high-probability event $\cup_{t \in [T]} \mathcal{E}^t_{\mathrm{opt}}$, we derive the regret bound from (8):

$$
\mathrm{Regret}(T) \leq \eta \sum_{t=1}^T \mathbb{E}_{x_t \sim d_0} \mathbb{E}_{a_t \sim \pi_t} \Big[
$$
$$
\big(\widehat{R}_{t-1}(x_t, a_t) + b_{t-1}(x_t, a_t) - R^*(x_t, a_t)\big)^2 \Big]
$$
$$
\leq 4\eta \sum_{t=1}^T \mathbb{E}_{x_t \sim d_0} \mathbb{E}_{a_t \sim \pi_t} \big(b_{t-1}(x_t, a_t)\big)^2,
$$

Thus, according to the definition of bonus $b_t$ in (3), we can follow standard analyses for general function approximation under the finite eluder dimension to obtain the desired result.

# 5. KL-Regularized RL

## 5.1. The Proposed Algorithm and Result

---
**Algorithm 2** KL-regularized LSVI with UCB
---
1: **Input:** $\eta, \epsilon, \pi_{\mathrm{ref}}, \mathcal{F}$.
2: **for** episode $t = 1, \ldots, T$ **do**
3:     Receive the intial state $s_1^t$.
4:     **for** stage $h = H, \ldots, 1$ **do**
5:         Set $\widehat{f}_h^t \leftarrow \mathrm{argmin}_{f_h \in \mathcal{F}_h} \sum_{i=1}^{t-1}(f_h(s_h^i, a_h^i) - r_h^i - \widehat{V}_{h+1}^t(s_{h+1}^i))^2$.
6:         Construct bonus $b_h^t(\cdot, \cdot)$ defined in (9).
7:         Set $\widehat{Q}_h^t(s, a) \leftarrow \widehat{f}_h^t(s, a) + b_h^t(s, a)$ for any $(s, a) \in \mathcal{S} \times \mathcal{A}$.
8:         Set $\widehat{V}_h^t(s) \leftarrow \max_\pi \mathbb{E}_{a \sim \pi(\cdot|s)}[\widehat{Q}_h^t(s, a) - \eta^{-1} \cdot \mathrm{KL}(\pi(\cdot|s), \pi_{\mathrm{ref}}(\cdot|s))]$.
9:     **end for**
10:     **for** stage $h = 1 \ldots, H$ **do**
11:         Observe $s_h^t$.
12:         Follow $\pi_h^t(\cdot|\cdot) \propto \pi_{\mathrm{ref}}(\cdot|\cdot) \exp[\eta \widehat{Q}_h^t(\cdot, \cdot)]$.
13:         Choose action $a_h^t \sim \pi_h^t(\cdot|s_h^t)$.
14:     **end for**
15: **end for**
---

We adapt the KL-regularization to the Least Square Value Iteration with UCB framework (Zhang, 2022; Wang et al., 2020) to propose the KL-LSVI-UCB algorithm in Algorithm 2, which establishes the optimistic $Q$-value function estimations $\widehat{Q}_h^t$ backward from step $H$ to step 1, with $\widehat{Q}_{H+1}^t \equiv 0$. In each step $h \in [H]$, we first learn the estimator $\widehat{f}_h^t$ to minimize the Bellman backward error:

$$
\widehat{f}_h^t = \mathrm{argmin}_{f_h \in \mathcal{F}_h} \sum_{i=1}^{t-1} (f_h(s_h^i, a_h^i) - r_h^i - \widehat{V}_{h+1}^t(s_{h+1}^i))^2.
$$

Then, under the principle of optimism, we construct the confidence set iteratively

$$
\mathcal{F}_h^{t-1} = \Big\{ f_h \in \mathcal{F}_h :
$$
$$
\sum_{i=1}^{t-1} (f_h(s_h^i, a_h^i) - \widehat{f}_h^t(s_h^i, a_h^i))^2 + \lambda \leq (\beta_h^T)^2 \Big\},
$$

and the bonus:

$$
b_h^t(s, a) = \min \Big\{ 1, \beta_h^T \cdot U_{\mathcal{F}_h^{t-1}}^h(\lambda, s, a; \mathcal{D}_{t-1}^h) \Big\}, \tag{9}
$$

where $U_{\mathcal{F}_h^{t-1}}^h$ is the uncertainty formulated in Definition 3.3, and $\mathcal{D}_{t-1}^h = \{s_h^i, a_h^i, r_h^i, s_{h+1}^i\}_{i \in [t-1]}$. Hence, we add the bonus to the value function estimation to get the optimistic estimation $\widehat{Q}_h^t(s, a) = \widehat{f}_h^t(s, a) + b_h^t(s, a)$ and

$$
\widehat{V}_h^t(s) = \frac{1}{\eta} \log \mathbb{E}_{a \sim \pi_{\mathrm{ref},h}(\cdot|s)} \exp\big(\eta \widehat{Q}_h^t(s, a)\big).
$$

For the analysis, we follow Ye et al. (2023); Wang et al. (2020) and assume that there exists a bonus function class

$\mathcal{B}_h$ with cardinality $N_{\mathcal{B}_h}$ that accommodates bonus functions $b_h^t$. Ye et al. (2023); Wang et al. (2020) have used examples and algorithms to show that this type of bonus can be approximately calculated under suitable conditions.

**Theorem 5.1.** *Under Algorithm 2 and Assumptions 3.6 and 3.7 with $\beta_h^T = \Theta(\sqrt{\log(N_h TH/\delta)})$, we have with probability at least $1 - \delta$,*

$$\text{Regret}(T) = \mathcal{O}\Big(\eta H^2 d(\mathcal{F}, \lambda, T) \cdot \log(N_{\mathcal{F} \oplus \mathcal{B}} TH/\delta)\Big),$$

*where $d(\mathcal{F}, \lambda, T) = \sum_{h=1}^H d(\mathcal{F}_h, \lambda, T)$ and $N_{\mathcal{F} \oplus \mathcal{B}} = \max_h N_{\mathcal{F}_h} \cdot N_{\mathcal{F}_{h+1}} \cdot N_{\mathcal{B}_{h+1}}$.*

### 5.2. Proof Outline of Theorem 5.1

We defer the detailed proof in Appendix B and highlight the crucial part of the proof in the sequel.

The crucial part of the analysis for the MDP setting lies in decomposing the regret into the errors of Bellman backup in each step. For conciseness, we omit the superscript $t$ for estimators when there is no confusion. We denote the Bellman error as $e_h(s_h, a_h) = \widehat{Q}_h(s_h, a_h) - \mathcal{T}_\eta^h \widehat{V}_{h+1}(s_h, a_h)$.

**Review of Standard Policy Loss Decomposition.** We first recall the standard decomposition without KL-regularization (Zhang, 2023):

$$\begin{aligned}
V_1^{\pi^*}(s_1) &- V_1^{\widehat{\pi}}(s_1) \leq \widehat{V}_1(s_1) - V_1^{\widehat{\pi}}(s_1) \\
&\leq \mathbb{E}^{\widehat{\pi}}\big[\widehat{Q}_1(s_1, a_1) - \mathcal{T}^1 \widehat{V}_2(s_1, a_1) \\
&\quad + \mathcal{T}^1 \widehat{V}_2(s_1, a_1) - \mathcal{T}^1 V_2^{\widehat{\pi}}(s_1, a_1)\big] \\
&\leq \cdots \leq \mathbb{E}^{\widehat{\pi}}\Big[\sum_{h=1}^H \big(\widehat{Q}_h(s_h, a_h) - \mathcal{T}_\eta^h \widehat{V}_{h+1}(s_h, a_h)\big)\Big],
\end{aligned}$$ (10)

where $\mathbb{E}^{\widehat{\pi}}$ denotes the expectation of trajectories generated by $d_0 \times \widehat{\pi}$, and $\mathcal{T}$ denotes the Bellman operator without KL-regularization. Thus, following the standard decomposition results in the polynomial dependence on $T$ for the previous works (Xiong et al., 2024a; Xie et al., 2024).

**Our Analysis.** In this work, to avoid the direct summation of Bellman errors as in (10), we develop a novel decomposition to obtain the square of the Bellman errors. To realize this goal, instead of decomposing the value functions, we decompose the policies.

For each $h$, let $\widehat{\pi}^{(h)} := \widehat{\pi}_{1:h} \oplus \pi_{h+1:H}^*$ be the concatenated policy of $\widehat{\pi}$ and $\pi^*$ at time step $h$. Then, $\pi^*$ can be denoted as $\widehat{\pi}^{(0)}$ and $V_1^* = V_1^{\widehat{\pi}^{(0)}}$. Then the suboptimality gap for $\widehat{\pi}$ can be decomposed as follows:

$$J(\pi^*) - J(\widehat{\pi}) = \sum_{h=0}^{H-1} \mathbb{E}_{s_1 \sim d_0} \underbrace{\big[V_1^{\widehat{\pi}^{(h)}}(s_1) - V_1^{\widehat{\pi}^{(h+1)}}(s_1)\big]}_{I_{h+1}}.$$

For each term $I_{h+1}$, since $\widehat{\pi}^{(h)}$ and $\widehat{\pi}^{(h+1)}$ only differs at step $h+1$, and they are both $\pi_{h'}^*$ for $h' = h+2, \ldots, H$, we use the notation $\text{KL}(\pi_h, \pi_h') := \text{KL}(\pi_h(\cdot|s_h) \| \pi_h'(\cdot|s_h))$ for short and can reduce the step-wise gap back into the bandit gap:

$$\begin{aligned}
\mathbb{E}_{s_{h+1} \sim d_{h+1}^{\widehat{\pi}}} &\Big\{ Q_{h+1}^{\pi^*}(s_{h+1}, \pi_{h+1}^*) - \eta^{-1} \text{KL}(\pi_{h+1}^* \| \pi_{\text{ref},h+1}) \\
&\quad - \big[Q_{h+1}^{\pi^*}(s_{h+1}, \widehat{\pi}_{h+1}) - \eta^{-1} \text{KL}(\widehat{\pi}_{h+1}, \pi_{\text{ref},h+1})\big] \Big\} \\
&\leq \eta \mathbb{E}^{\widehat{\pi}}\big[\big(\widehat{Q}_{h+1}(s_{h+1}, a_{h+1}) - Q_{h+1}^*(s_{h+1}, a_{h+1})\big)^2\big],
\end{aligned}$$

where the inequality holds by applying (8) with $R^* = Q_{h+1}^{\pi^*}$ and $\widehat{R}_t + b_t = \widehat{Q}_{h+1}$. Therefore, we can obtain the following bound for each ($I_{h+1}$).

**Lemma 5.2.** *We use the notation $\mathbb{E}^{\widehat{\pi}}$ for the expectation of trajectories generated by $d_0 \times \widehat{\pi}$. Conditioning on the uniform optimism event that $\mathcal{E}_{\text{opt}} = \{\widehat{Q}(s_h, a_h) - Q_h^*(s_h, a_h) \geq 0, \ \forall(s_h, a_h) \in \mathcal{S} \times \mathcal{A}, \ h \in [H]\}$ holds,*

$$I_{h+1} \leq \eta \mathbb{E}_{\cdot|s_{h+1}, a_{h+1}}^{\widehat{\pi}} \Big[\Big(\sum_{j=h+1}^H e_j(s_j, a_j)\Big)^2\Big].$$

The proof is shown in Appendix B.1. Therefore, we can decompose the regret as follows:

$$\begin{aligned}
J(\pi^*) - J(\widehat{\pi}) &\leq \eta \sum_{h=0}^{H-1} \mathbb{E}^{\widehat{\pi}}\Big[\Big(\sum_{j=h+1}^H e_j(s_j, a_j)\Big)^2\Big] \\
&\leq \eta H^2 \mathbb{E}^{\widehat{\pi}}\Big[\sum_{h=1}^H (e_h(s_h, a_h))^2\Big],
\end{aligned}$$

where the second inequality is the Cauchy-Schwarz inequality. Since the summation over $H$ steps is inside the square, we can only pay an additional order of $H$.

Then, following the standard analysis of MDP literature (Zhang, 2023), we can demonstrate that the optimism $\mathcal{E}_{\text{opt}}$ holds with a high probability and obtain the desired bound.

## 6. Conclusion

In this paper, we study KL-regularized contextual bandits and MDPs in the standard online RL setting. While these frameworks have been widely applied in the empirical post-training of modern foundation generative models, their theoretical properties are substantially less explored. We propose two provably efficient algorithms: KL-regularized UCB and KL-regularized LSVI with UCB, based on the standard optimism principle and show that they achieve regret bounds that scale logarithmically with $T$, significantly improving over the typical $\mathcal{O}(\sqrt{T})$ rate. To the best of our knowledge, this is the first theoretical analysis of KL-regularized RL to establish such logarithmic regret bounds in the standard online setting, aligning well with the empirically observed

superior sample efficiency of KL-regularized RL (Bai et al., 2022; DeepSeek-AI et al., 2025). An expense of this logarithmic bound for MDPs is that the bound has additional dependence on the horizon $H$, which can be left as future work.

The key to this improvement lies in a refined value decomposition for the bandit setting and a novel policy decomposition technique for MDPs, both of which may be of independent interest. We hope our study inspires further theoretical investigations into the learning dynamics of KL-regularized RL.

## Impact Statement

This paper presents work whose goal is to advance the field of Machine Learning. There are many potential societal consequences of our work, none of which we feel must be specifically highlighted here.

## Acknowledgements

We thank the anonymous reviewers for their helpful comments. HZ and QG are supported in part by the National Science Foundation DMS-2323113 and IIS-2403400. HZ is also partially supported by the Amazon PhD fellowship. The views and conclusions contained in this paper are those of the authors and should not be interpreted as representing any funding agencies.

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

## A. Proofs for KL-Regularized Contextual Bandits

**Lemma A.1** (Objective Decomposition). *For any $t \in [T]$, conditioning on the uniform optimism event that $\mathcal{E}^t_{\mathrm{opt}} = \{\widehat{R}_t(x, a) + b_t(x, a) - R^*(x, a) \geq 0, \ \forall (x, a) \in \mathcal{X} \times \mathcal{A}\}$ holds, we have*

$$J(\pi^*) - J(\pi_t) \leq \eta \mathbb{E}_{x \sim d_0} \mathbb{E}_{a \sim \pi_t} \left[ \left( \widehat{R}_{t-1}(x, a) + b_{t-1}(x, a) - R^*(x, a) \right)^2 \right].$$

*Proof.* For all $t \geq 1$, with $R^*(x, a) = \frac{1}{\eta} \log \exp \left( \eta R^*(x, a) \right)$ and the closed-form solution of $\pi^*$ and $\pi$ from Lemma 3.4, we have the following equality for the suboptimality gap:

$$\mathbb{E}_{\pi^*} \left[ R^*(x, a) - \frac{1}{\eta} \log \frac{\pi^*(a|x)}{\pi_{\mathrm{ref}}(a|x)} \right] - \mathbb{E}_{\pi_t} \left[ R^*(x, a) - \frac{1}{\eta} \log \frac{\pi_t(a|x)}{\pi_{\mathrm{ref}}(a|x)} \right]$$

$$= \frac{1}{\eta} \mathbb{E}_{\pi^*} \left[ \log \frac{\pi_{\mathrm{ref}}(a|x) \cdot \exp(\eta R^*(x, a))}{\pi^*(a|x)} \right] - \frac{1}{\eta} \mathbb{E}_{\pi_t} \left[ \log \frac{\pi_{\mathrm{ref}}(a|x) \cdot \exp(\eta R^*(x, a))}{\pi_t(a|x)} \right]$$

$$= \frac{1}{\eta} \mathbb{E}_{\pi^*} \left[ \log \frac{\pi_{\mathrm{ref}}(a|x) \cdot \exp(\eta R^*(x, a))}{\pi^*(a|x)} \right] - \frac{1}{\eta} \mathbb{E}_{\pi_t} \left[ \log \frac{\pi_{\mathrm{ref}}(a|x) \cdot \exp(\eta (\widehat{R}_{t-1}(x, a) + b_{t-1}(x, a)))}{\pi_t(a|x)} \right]$$

$$\quad + \mathbb{E}_{\pi_t} \left[ \widehat{R}_{t-1}(x, a) - R^*(x, a) + b_{t-1}(x, a) \right]$$

$$= \frac{1}{\eta} \log Z_{R^*}(x) - \frac{1}{\eta} \log Z_{\widehat{R}_{t-1}+b_{t-1}}(x) + \mathbb{E}_{\pi_t} \left[ \widehat{R}_{t-1}(x, a) - R^*(x, a) + b_{t-1}(x, a) \right], \tag{11}$$

where $Z_R(x) := \sum_{a \in \mathcal{A}} \pi_{\mathrm{ref}}(a|x) \cdot \exp(\eta R(x, a))$ is the normalization term for the policy with respect to reward function $R$.

We define $\Delta(x, R)$ as the following quantity for a reward function $R : \mathcal{X} \times \mathcal{A} \to \mathbb{R}$:

$$\Delta(x, R) := -\frac{1}{\eta} \log Z_R(x) + \sum_{a \in \mathcal{A}} \frac{\pi_{\mathrm{ref}}(a|x) \cdot \exp(\eta R(x, a))}{Z_R(x)} \cdot \left( R(x, a) - R^*(x, a) \right)$$

$$= -\frac{1}{\eta} \log Z_R(x) + \mathbb{E}_{\pi_R^\eta} \left[ R(x, a) - R^*(x, a) \right]. \tag{12}$$

With the definition of $\Delta(x, a)$, the gap in (11) can be written as $\mathbb{E}_{x \sim d_0} [\Delta(x, \widehat{R}_{t-1} + b_{t-1}) - \Delta(x, R^*)]$. To analyze the behavior of $\mathbb{E}_{x \sim d_0} [\Delta(x, \widehat{R}_{t-1} + b_{t-1}) - \Delta(x, R^*)]$, we now study the gradient of $\Delta(x, R)$. We are interested in the gradient of $\Delta(x, R)$ with respect to $R(x, a)$ for each $a \in \mathcal{A}$.

Then, by invoking Lemma A.2 to simplify the computation, we can write the gradient as follows:

$$\frac{\partial \Delta(x, R)}{\partial R(x, a)} = \frac{\partial \left[ -\frac{1}{\eta} \log Z_R(x) + \mathbb{E}_{\pi_R^\eta} \left[ R(x, a) - R^*(x, a) \right] \right]}{\partial R(x, a)}$$

$$= -\frac{1}{\eta} \frac{1}{Z_R(x)} \frac{\partial Z_R(x)}{\partial R(x, a)} + \frac{\partial}{\partial R(x, a)} \left[ \pi_R^\eta(a|x) \cdot \left( R(x, a) - R^*(x, a) \right) \right]$$

$$\quad + \frac{\partial}{\partial R(x, a)} \left[ \sum_{a' \neq a} \pi_R^\eta(a'|x) \cdot \left( R(x, a') - R^*(x, a') \right) \right]$$

$$= -\pi_R^\eta(a|x) + \left[ \pi_R^\eta(a|x) + \frac{\partial \pi_R^\eta(a|x)}{\partial R(x, a)} \cdot \left( R(x, a') - R^*(x, a') \right) \right]$$

$$\quad + \sum_{a' \neq a} \frac{\partial \pi_R^\eta(a'|x)}{\partial R(x, a)} \cdot \left( R(x, a) - R^*(x, a) \right)$$

$$= \left[ \eta \pi_R^\eta(a|x) - \eta \pi_R^\eta(a|x)^2 \right] \cdot \left( R(x, a) - R^*(x, a) \right) - \sum_{a' \neq a} \eta \pi_R^\eta(a|x) \cdot \pi_R^\eta(a'|x) \cdot \left( R(x, a') - R^*(x, a') \right)$$

$$= \eta \pi_R^\eta(a|x) \cdot \left( R(x, a) - R^*(x, a) \right) - \eta \sum_{a' \in \mathcal{A}} \pi_R^\eta(a|x) \cdot \pi_R^\eta(a'|x) \cdot \left( R(x, a') - R^*(x, a') \right),$$

where the third equality uses (14) and the fourth equality uses (15) and (16), the second equality and the last equality simply use the linearity of $\partial$ and summation.

Since $\frac{\partial \Delta(x,R)}{\partial R(x,a)}$ is uniformly bounded when $R$ is uniformly bounded, by the Mean Value Theorem, there exists $f_\lambda = \lambda \cdot (\widehat{R}_{t-1} + b_{t-1}) + (1-\lambda) \cdot R^*$ for some $\lambda \in [0,1]$ such that

$$
\mathbb{E}_{x \sim d_0}[\Delta(x, \widehat{R}_{t-1} + b_{t-1}) - \Delta(x, R^*)]
$$

$$
= \mathbb{E}_{x' \sim d_0}\Big[ \sum_{a \in \mathcal{A}} \frac{\partial \Delta(x', f_\lambda)}{\partial R(x,a)}(\widehat{R}_{t-1}(x,a) + b_{t-1}(x,a) - R^*(x,a)) \Big]
$$

$$
= \mathbb{E}_{x' \sim d_0}\Big[ \sum_{a \in \mathcal{A}} \Big(\eta \pi_{f_\lambda}^\eta(a|x) \cdot \big(f_\lambda(x,a) - R^*(x,a)\big) - \eta \sum_{a' \in \mathcal{A}} \pi_{f_\lambda}^\eta(a|x) \cdot \pi_{f_\lambda}^\eta(a'|x) \cdot \big(f_\lambda(x,a') - R^*(x,a')\big)\Big)
$$

$$
\cdot (\widehat{R}_{t-1}(x,a) + b_{t-1}(x,a) - R^*(x,a)) \Big]
$$

$$
\leq \eta \mathbb{E}_{x \sim d_0}\Big[ \sum_{a \in \mathcal{A}} \big(\widehat{R}_{t-1}(x,a) + b_{t-1}(x,a) - R^*(x,a)\big) \cdot \underbrace{\big(f_\lambda(x,a) - R^*(x,a)\big) \cdot \pi_{f_\lambda}^\eta(a|x)}_{U(x,a)} \Big], \tag{13}
$$

where the last inequality holds since conditioning on $\mathcal{E}_{\mathrm{opt}}^t$ we have

$$
f_\lambda(x,a') - R^*(x,a') = \lambda(\widehat{R}_{t-1}(x,a) + b_{t-1}(x,a) - R^*(x,a))) \geq 0, \quad \forall (x,a) \in \mathcal{X} \times \mathcal{A}.
$$

We use the notation $\delta_t = \widehat{R}_t + b_t - R^*$. Let

$$
U(\lambda) = \sum_{a \in \mathcal{A}} \lambda \delta_{t-1}(x,a)^2 \cdot \frac{\pi_0(a|x) \cdot \exp\big(\eta(\lambda \delta_{t-1}(x,a) + R^*(x,a))\big)}{\mathbb{E}_{\pi_0} \exp\big(\eta(\lambda \delta_{t-1}(x,a) + R^*(x,a))\big)}, \quad \text{for } \lambda \in [0,1].
$$

We take the derivative as follows:

$$
\frac{\partial U(\lambda)}{\partial \lambda} = \sum_{a \in \mathcal{A}} \Big[ \delta_{t-1}(x,a)^2 \pi_{f_\lambda}^\eta + \lambda \delta_{t-1}(x,a)^2 \cdot \Big(\frac{\pi_0(a|x) \exp\big(\eta(\lambda \delta_{t-1}(x,a) + R^*(x,a))\big) \cdot \eta \delta_{t-1}(x,a)}{\mathbb{E}_{\pi_0} \exp\big(\eta(\lambda \delta_{t-1}(x,a) + R^*(x,a))\big)}\Big)
$$

$$
- \frac{\pi_0(a|x) \exp\big(\eta(\lambda \delta_{t-1}(x,a) + R^*(x,a))\big) \cdot \mathbb{E}_{\pi_0} \eta \delta_{t-1}(x,a) \cdot \exp\big(\eta(\lambda \delta_{t-1}(x,a) + R^*(x,a))\big)}{\big(\mathbb{E}_{\pi_0} \exp\big(\eta(\lambda \delta_{t-1}(x,a) + R^*(x,a))\big)\big)^2} \Big]
$$

$$
= \sum_{a \in \mathcal{A}} \Big[ \delta_{t-1}(x,a)^2 \pi_{f_\lambda}^\eta + \lambda \delta_{t-1}(x,a)^2 \pi_{f_\lambda}^\eta \cdot \eta\big(\delta_{t-1}(x,a) - \mathbb{E}_{\pi_{f_\lambda}^\eta} \delta_{t-1}(x,a)\big) \Big]
$$

$$
= \mathbb{E}_{\pi_{f_\lambda}^\eta}\big[\delta_{t-1}(x,a)^2\big] + \lambda\Big\{ \mathbb{E}_{\pi_{f_\lambda}^\eta}\big[\delta_{t-1}(x,a)^3\big] - \mathbb{E}_{\pi_{f_\lambda}^\eta}\big[\delta_{t-1}(x,a)^2\big] \cdot \mathbb{E}_{\pi_{f_\lambda}^\eta}\big[\delta_{t-1}(x,a)\big] \Big\}
$$

$$
\geq 0,
$$

where the inequality holds since for random variable $X \geq 0$ we have

$$
\mathbb{E}[X^3] - \mathbb{E}[X^2] \cdot \mathbb{E}[X] = \mathbb{E}[(X^2 - \mathbb{E}[X^2])(X - \mathbb{E}[X])]
$$

$$
= \mathbb{E}\big[\big(X^2 - (\mathbb{E}[X])^2\big)(X - \mathbb{E}[X])\big] + \mathbb{E}\big[\big((\mathbb{E}[X])^2 - \mathbb{E}[X^2]\big)(X - \mathbb{E}[X])\big]
$$

$$
= \mathbb{E}\big[(X + \mathbb{E}[X])(X - \mathbb{E}[X])^2\big] + 0
$$

$$
\geq 0,
$$

, which follows from Lemma 2.14 in (Zhao et al., 2025). Therefore, since $f_\lambda(x,a) \leq \widehat{R}_{t-1}(x,a) + b_{t-1}(x,a)$ for any $(x,a) \in \mathcal{X} \times \mathcal{A}$, we can obtain that

$$
U(\lambda) \leq U(1) = \sum_{a \in \mathcal{A}} \delta_{t-1}(x,a)^2 \cdot \pi_{\widehat{R}_{t-1} + b_{t-1}}^\eta(a|x).
$$

We further have the following bound for RHS of (13):

$$
J(\pi^*) - J(\pi_t) = \mathbb{E}_{x \sim d_0}[\Delta(x, \widehat{R}_{t-1} + b_{t-1}) - \Delta(x, R^*)]
$$

$$\leq \eta \mathbb{E}_{x \sim d_0} \mathbb{E}_{a \sim \pi^{\eta}_{\widehat{R}_{t-1}+b_{t-1}}(\cdot|x)} \left[ \left( \widehat{R}_{t-1}(x,a) + b_{t-1}(x,a) - R^*(x,a) \right)^2 \right].$$

$\square$

**Lemma A.2.** *Let* $Z_R(x) := \sum_{a \in \mathcal{A}} \pi_{\text{ref}}(a|x) \cdot \exp(\eta R(x,a))$ *and* $\pi^{\eta}_R(a|x) = \frac{\pi_{\text{ref}}(a|x)\exp(\eta R(x,a))}{Z_R(x)}$. *We have the following partial gradient results:*

$$\frac{\partial Z_R(x)}{\partial R(x,a)} = \eta \pi_{\text{ref}}(a|x) \exp\left( \eta R(x,a) \right), \tag{14}$$

$$\frac{\partial \pi^{\eta}_R(a|x)}{\partial R(x,a)} = \eta \pi^{\eta}_R(a|x) - \eta \pi^{\eta}_R(a|x)^2. \tag{15}$$

*For* $a' \neq a$, *we have*

$$\frac{\partial \pi^{\eta}_R(a'|x)}{\partial R(x,a)} = -\eta \pi^{\eta}_R(a|x) \cdot \pi^{\eta}_R(a'|x). \tag{16}$$

*Proof.* The proof is deferred to Appendix C. $\square$

**Lemma A.3.** *Under Algorithm 1 and Assumption 3.2 and the condition that the noises* $\epsilon_t$ *are conditional* 1*-sub-Gaussian, we have with probability at least* $1-\delta$ *for all* $t \in [T]$, *the uniform optimism event that* $\mathcal{E}^t_{\text{opt}} = \{\widehat{R}_t(x,a) + b_t(x,a) - R^*(x,a) \geq 0, \ \forall (x,a) \in \mathcal{X} \times \mathcal{A}\}$ *holds true.*

*Proof.* By invoking Lemma C.1 with $\mathcal{F} = \mathcal{R}$, $\mathcal{Z} = \mathcal{X} \times \mathcal{A}$ and $\eta = 1$, we have for all $t \in [T]$ with probability at least $1 - \delta$,

$$\sum_{i=1}^{t} (\widehat{R}_t(x_i,a_i) - R^*(x_i,a_i))^2 \leq 8 \log(N_{\mathcal{R}} T/\delta) = \frac{1}{2}\beta_T^2. \tag{17}$$

Hence, we deduce that for any $(x,a) \in \mathcal{X} \times \mathcal{A}$,

$$R^*(x,a) - \widehat{R}_t(x,a) \leq \sup_{R_1,R_2 \in \mathcal{R}} \frac{|R_1(x,a) - R_2(x,a)|}{\sqrt{\lambda + \sum_{i=1}^{t}(R_1(x_i,a_i) - R_2(x_i,a_i))^2}} \cdot \sqrt{\lambda + \sum_{i=1}^{t}(R^*(x_i,a_i) - \widehat{R}_t(x_i,a_i))^2}$$

$$\leq U_{\mathcal{R}_t}(\lambda, x, a; \mathcal{D}_t) \cdot \sqrt{\lambda + \frac{1}{2}\beta_T^2}$$

$$\leq U_{\mathcal{R}_t}(\lambda, x, a; \mathcal{D}_t) \cdot \beta_T,$$

where the second inequality follows from the definition of $U_{\mathcal{R}}$ and (17), and the last inequality uses $\lambda \leq \frac{1}{2}\beta_T^2$ and the choice of bonus $b_t$. Besides, since $R^*(x,a) - \widehat{R}_t(x,a) \leq 1$, we have

$$R^*(x,a) - \widehat{R}_t(x,a) \leq b_t(x,a).$$

Therefore, we get

$$\widehat{R}_t(x,a) + b_t(x,a) - R^*(x,a) \geq 0.$$

$\square$

*Proof of Theorem 4.1.* Let

$$U_t := \sup_{R_1,R_2 \in \mathcal{R}_{t-1}} \frac{|R_1(x_t,a_t) - R_2(x_t,a_t)|}{\sqrt{\lambda + \sum_{i=1}^{t-1}(R_1(x_i,a_i) - R_2(x_i,a_i))^2}}.$$

We know from Lemma A.3 that $\cup_{t \in [T]} \mathcal{E}^t_{\text{opt}}$ holds with probability at least $1 - \delta$. Thus, conditioning on $\cup_{t \in [T]} \mathcal{E}^t_{\text{opt}}$, we invoke Lemma A.1 to get

$$\text{Regret}(T) = \sum_{t=1}^{T} (J(\pi^*) - J(\pi_t))$$

$$\leq \eta \sum_{t=1}^{T} \mathbb{E}_{x_t \sim d_0} \mathbb{E}_{a_t \sim \pi_t} \left[ \left( \widehat{R}_{t-1}(x_t, a_t) + b_{t-1}(x_t, a_t) - R^*(x_t, a_t) \right)^2 \right]$$

$$\leq 4\eta \sum_{t=1}^{T} \mathbb{E}_{x_t \sim d_0} \mathbb{E}_{a_t \sim \pi_t} \left( b_{t-1}(x_t, a_t) \right)^2$$

$$= 4\eta \beta_T^2 \sum_{t=1}^{T} \mathbb{E}_{x_t \sim d_0} \mathbb{E}_{a_t \sim \pi_t} \min \left\{ 1, \left( U_{\mathcal{R}_{t-1}}(\lambda, x_t, a_t; \mathcal{D}_{t-1}) \right)^2 \right\}$$

$$\leq \mathcal{O} \left( \eta \log(N_{\mathcal{R}} T / \delta) \cdot d(\mathcal{R}, \lambda, T) \right),$$

where the second inequality uses the condition $\mathcal{E}_{t-1}$, the second equality uses the definition of bonus $b_t$ in (3), and the last inequality follows from the value of $\beta_T$ and the eluder dimension in Definition 3.3. □

## B. Proofs for KL-Regularized RL

### B.1. Regret Decomposition

**Lemma B.1.** *We use the short-hand notation $\mathbb{E}^{\widehat{\pi}}$ for the expectation of trajectories generated by $d_0 \times \widehat{\pi}$, and*

$$e_h(s_h, a_h) = \widehat{Q}_h(s_h, a_h) - \mathcal{T}_\eta^h \widehat{V}_{h+1}(s_h, a_h) = \widehat{Q}_h(s_h, a_h) - \mathbb{E}_{r_h, s_{h+1} | s_h, a_h} [r_h + \widehat{V}_{h+1}(s_{h+1})].$$

*Conditioning on the uniform optimism event that $\mathcal{E}_{\text{opt}} = \{ \widehat{Q}(s_h, a_h) - Q_h^*(s_h, a_h) \geq 0, \ \forall (s_h, a_h) \in \mathcal{S} \times \mathcal{A}, \ h \in [H] \}$ holds, we have*

$$J(\pi^*) - J(\widehat{\pi}) \leq \eta \mathbb{E}^{\widehat{\pi}} \sum_{h=1}^{H} \left[ \left( \sum_{j=h}^{H} e_j(s_j, a_j) \right) \right)^2 \right].$$

*Proof.* For each $h$, let $\widehat{\pi}^{(h)} := \widehat{\pi}_{1:h} \oplus \pi_{h+1:H}^*$ be the concatenated policy of $\widehat{\pi}$ and $\pi^*$ at time step $h$. Then, $\pi^*$ can be denoted as $\widehat{\pi}^{(0)}$ and $V_1^* = V_1^{\widehat{\pi}^{(0)}}$. Then the suboptimality gap for $\widehat{\pi}$ can be decomposed as follows:

$$J(\pi^*) - J(\widehat{\pi}) = V_1^{\pi^*}(s_1) - V_1^{\widehat{\pi}}(s_1) = \sum_{h=0}^{H-1} \mathbb{E}_{s_1 \sim d_0} \underbrace{\left[ V_1^{\widehat{\pi}^{(h)}}(s_1) - V_1^{\widehat{\pi}^{(h+1)}}(s_1) \right]}_{I_{h+1}}.$$

For each term $I_{h+1}$, we deduce that

$$\mathbb{E}_{s_1 \sim d_0, s_{h+1} \sim d_{h+1}^{\widehat{\pi}}} \left[ V_{h+1}^{\widehat{\pi}^{(h)}}(s_{h+1}) - V_{h+1}^{\widehat{\pi}^{(h+1)}}(s_{h+1}) \right]$$

$$= \mathbb{E}_{s_1 \sim d_0, s_{h+1} \sim d_{h+1}^{\widehat{\pi}}} \left\{ \mathbb{E}_{\pi_{h+1}^*} \left[ Q_{h+1}^{\widehat{\pi}^{(h)}}(s_{h+1}, a_{h+1}) - \eta^{-1} \text{KL}(\pi_{h+1}^*(\cdot | s_{h+1}) \| \pi_{\text{ref}, h+1}(\cdot | s_{h+1})) \right] \right.$$

$$\left. - \mathbb{E}_{\widehat{\pi}_{h+1}} \left[ Q_{h+1}^{\widehat{\pi}^{(h+1)}}(s_{h+1}, a_{h+1}) - \eta^{-1} \text{KL}(\widehat{\pi}_{h+1}(\cdot | s_{h+1}) \| \pi_{\text{ref}, h+1}(\cdot | s_{h+1})) \right] \right\}$$

$$= \mathbb{E}_{s_1 \sim d_0, s_{h+1} \sim d_{h+1}^{\widehat{\pi}}} \left\{ \mathbb{E}_{\pi_{h+1}^*} \left[ Q_{h+1}^{\pi^*}(s_{h+1}, a_{h+1}) - \eta^{-1} \text{KL}(\pi_{h+1}^*(\cdot | s_{h+1}) \| \pi_{\text{ref}, h+1}(\cdot | s_{h+1})) \right] \right.$$

$$\left. - \mathbb{E}_{\widehat{\pi}_{h+1}} \left[ Q_{h+1}^{\pi^*}(s_{h+1}, a_{h+1}) - \eta^{-1} \text{KL}(\widehat{\pi}_{h+1}(\cdot | s_{h+1}) \| \pi_{\text{ref}, h+1}(\cdot | s_{h+1})) \right] \right\},$$

where the second equality follows from the fact that $\widehat{\pi}_l^{(h)} = \widehat{\pi}_l^{(h+1)} = \pi_h^*$ for $l = h + 2, \dots, H$.

Then, we can follow the analysis of Lemma A.1 for each step $h + 1$ with $d_0 = d_{h+1}^{\widehat{\pi}}$, $R^* = Q_{h+1}^{\pi^*}$, $\pi^* = \pi_{h+1}^*$ and $\widehat{\pi} = \widehat{\pi}_{h+1}$ to derive that

$$I_{h+1} \leq \eta \mathbb{E}^{\widehat{\pi}} \left[ \left( \widehat{Q}_{h+1}(s_{h+1}, a_{h+1}) - Q_{h+1}^*(s_{h+1}, a_{h+1}) \right)^2 \right].$$

Further, conditioning on $\mathcal{E}_{\text{opt}}$, we deduce that

$$0 \geq Q_{h+1}^*(s_{h+1}, a_{h+1}) - \widehat{Q}_{h+1}(s_{h+1}, a_{h+1})$$

$$= \mathbb{E}_{s_{h+2}|s_{h+1},a_{h+1}}(V_{h+2}^*(s_{h+2}) - \widehat{V}_{h+2}(s_{h+2})) - e_{h+1}(s_{h+1}, a_{h+1})$$

$$= \mathbb{E}_{\cdot|s_{h+1},a_{h+1}}^{\widehat{\pi}} \eta^{-1} \log \mathbb{E}_{a_{h+2}\sim\pi_{\text{ref},h+2}} e^{\eta(Q_{h+2}^*(s_{h+2},a_{h+2}) - \widehat{Q}_{h+2}(s_{h+2},a_{h+2}))} - e_{h+1}(s_{h+1}, a_{h+1})$$

$$\geq \mathbb{E}_{\cdot|s_{h+1},a_{h+1}}^{\widehat{\pi}} \Big[ \big( Q_{h+2}^*(s_{h+2},a_{h+2}) - \widehat{Q}_{h+2}(s_{h+2},a_{h+2}) \big) - e(s_{h+1},a_{h+1}) \Big]$$

$$\geq \cdots$$

$$\geq -\mathbb{E}_{\cdot|s_{h+1},a_{h+1}}^{\widehat{\pi}} \sum_{j\geq h+1} e_j(s_j, a_j),$$

where second inequality holds due to the Jensen's inequality. Taking this back to $I_{h+1}$ leads to

$$I_{h+1} \leq \eta \mathbb{E}^{\widehat{\pi}} \Big[ \big( \mathbb{E}_{\cdot|s_{h+1},a_{h+1}}^{\widehat{\pi}} \sum_{j\geq h+1} e_j(s_j, a_j) \big)^2 \Big]$$

$$\leq \eta \mathbb{E}_{\cdot|s_{h+1},a_{h+1}}^{\widehat{\pi}} \Big[ \big( \sum_{j=h+1}^{H} e_j(s_j, a_j) \big)^2 \Big],$$

where the second inequality uses the Jensen's inequality.

Therefore, we get

$$J(\pi^*) - J(\widehat{\pi}) \leq \eta \sum_{h=0}^{H-1} \mathbb{E}^{\widehat{\pi}} \Big[ \big( \sum_{j=h+1}^{H} e_j(s_j, a_j) \big)^2 \Big].$$

$\square$

**Lemma B.2** (Confidence Sets for Value Function). *Let $\widehat{f}_h^t$ be the estimator as defined in Algorithm 2 and*

$$\beta_h^T = 4\sqrt{\log(4N_h TH/\delta)},$$

*where we define $N_h = N_{\mathcal{F}_h} \cdot N_{\mathcal{F}_{h+1}} \cdot N_{\mathcal{B}_{h+1}(\lambda)}$. Then, with probability at least $1 - \delta$, we have for all $t \in [T]$ and $h \in [H]$,*

$$|\widehat{f}_h^t(s,a) - \mathcal{T}_\eta^h \widehat{V}_{h+1}^t(s,a)| \leq \beta_h^T \cdot U_{\mathcal{F}_h}^h(s,a; \mathcal{D}_{t-1}^h) \quad \forall(s,a) \in \mathcal{S} \times \mathcal{A},$$

$$\widehat{Q}_h^t(s_h, a_h) - Q_h^*(s_h, a_h) \geq 0,$$

*where $\mathcal{D}_{t-1}^h$ denotes the history data $\{x_h^i, a_h^i\}_{i\in[t-1]}$.*

*Proof.* For $k \in [K]$, $h \in [H]$, let $\mathcal{S}_h^t$ be the $\sigma$-algebra generated by the random variables representing the state-action pairs up to and including those that appear stage $h$ of episode $k$. That is, $\mathcal{S}_h^t$ is generated by

$$s_1^1, a_1^1, \ldots, s_h^1, a_h^1, \ldots, s_H^1, a_H^1,$$
$$s_1^2, a_1^2, \ldots, s_h^2, a_h^2, \ldots, s_H^2, a_H^2,$$
$$\vdots$$
$$s_1^t, a_1^t, \ldots, s_h^t, a_h^t.$$

Let $\mathcal{Q}_{h+1}, V_{h+1}$ be the set of possible value functions, i.e.

$$\mathcal{Q}_{h+1} = \big\{ g : \mathcal{S} \times \mathcal{A} \to \mathbb{R} | \exists f_{h+1} \in \mathcal{F}_{h+1}, b_{h+1} \in \mathcal{B}_{h+1} \text{ s.t. } g = f_{h+1} + b_{h+1} \big\},$$
$$\mathcal{V}_{h+1} = \big\{ g : \mathcal{S} \to \mathbb{R} | \exists Q_{h+1} \in \mathcal{Q}_{h+1} \text{ s.t. } g(\cdot) = \max_{a\in\mathcal{A}} Q_{h+1}(\cdot, a) \big\}.$$

The proof is based on the standard concentration inequality for the least square estimator (Zhang, 2023).

For simplicity, we denote by $e_h^t$ the noise term in the least square estimator with respect to an arbitrary value function $V_{h+1} \in \mathcal{V}_{h+1}$, i.e.,

$$e_h^t := r_h^t + V_{h+1}(s_{h+1}^t) - \mathcal{T}_\eta^h V_{h+1}(s_h^t, a_h^t).$$

By definition,

$$\mathbb{E}[e_h^t | \mathcal{S}_h^t] = 0, \quad |e_h^t| \leq 1.$$

Hence, by invoking Lemma C.1 and using a union bound over $\mathcal{V}_{h+1}$, we have for all $t \in [T]$ and $h \in [H]$, with probability at least $1 - \delta$,

$$\sum_{i=1}^{t-1} \left[ \widehat{f}_h^t(s_h^i, a_h^i) - \mathcal{T}_\eta^h \widehat{V}_{h+1}^t(s_h^i, a_h^i) \right]^2 \leq 8 \log(4N_h t H/\delta) \leq \frac{1}{2}(\beta_h^T)^2, \tag{18}$$

where the first inequality follows from the definition of $N_h$ and $|\mathcal{V}_{h+1}| \leq N_{\mathcal{F}_{h+1}} \cdot N_{\mathcal{B}_{h+1}}$. Then we can complete the proof using union bound over all $t \geq 1$.

Therefore, for any $(s, a) \in \mathcal{S} \times \mathcal{A}$, we have

$$|\widehat{f}_h^t(s, a) - \mathcal{T}_\eta^h \widehat{V}_{h+1}^t(s, a)| \leq U_{\mathcal{F}_h}^h(s, a; \mathcal{D}_{t-1}^h) \cdot \sqrt{\lambda + \sum_{i=1}^{t-1} \left[ \widehat{f}_h^t(s_h^i, a_h^i) - \mathcal{T}_\eta^h \widehat{V}_{h+1}^t(s_h^i, a_h^i) \right]^2}$$

$$\leq (\beta_h^T)^2 \cdot U_{\mathcal{F}_h}^h(s, a; \mathcal{D}_{t-1}^h), \tag{19}$$

where the first inequality holds by using the completeness of the Bellman operator $\mathcal{T}_\eta^h \widehat{V}_{h+1}^t \in \mathcal{F}_h$ and the definition of uncertainty $U_{\mathcal{F}_h}^h$, and the last inequality applies (18) and $\lambda \leq \frac{1}{2}(\beta_h^T)^2$.

Moreover, we will prove $\widehat{Q}_h^t(s_h, a_h) - Q_h^*(s_h, a_h) \geq 0, \forall h \in [H]$ by induction. At step $H + 1$, we know that $Q_h^* = \widehat{Q} \equiv 0$. Assume that at step $h + 1$, we have

$$\widehat{Q}_{h+1}^t(s_{h+1}, a_{h+1}) - Q_{h+1}^*(s_{h+1}, a_{h+1}) \geq 0.$$

Then, at step $h$, we obtain that

$$\begin{aligned}
\widehat{Q}_h^t(s_h, a_h) - Q_h^*(s_h, a_h) &= \widehat{f}_h^t(s_h, a_h) + b_h^t(s_h, a_h) - \mathcal{T}_\eta^h V_{h+1}^*(s_h, a_h) \\
&\geq \mathcal{T}_\eta^h \widehat{V}_{h+1}^t(s_h, a_h) - (\beta_h^T)^2 \cdot U_{\mathcal{F}_h}^h(s, a; \mathcal{D}_{t-1}^h) + (\beta_h^T)^2 \cdot U_{\mathcal{F}_h}^h(s, a; \mathcal{D}_{t-1}^h) - \mathcal{T}_\eta^h V_{h+1}^*(s_h, a_h) \\
&\geq \frac{1}{\eta} \mathbb{E}_{s_{h+1} | s_h, a_h} \left[ \log \mathbb{E}_{a_{h+1} \sim \pi_{\text{ref}, h+1}(\cdot | s_{h+1})} \exp\left( \eta(\widehat{Q}_{h+1}^t(s_{h+1}, a_{h+1}) - Q_{h+1}^*(s_{h+1}, a_{h+1}))) \right) \right] \\
&\geq 0,
\end{aligned}$$

where the first inequality uses (19) and the definition of bonus $b_h^t$ in (3), the second inequality follows from the derivation of the value function in (1), and the last inequality is due to the induction hypothesis at step $h + 1$. □

*Proof of Theorem 5.1.* From Lemma B.2 we know that $\mathcal{E}_{\text{opt}}$ holds at all time step $t \in [T]$ with probability a least $1 - \delta$. Hence, by applying Lemma 5.2 for each $t \in [T]$, we have

$$\begin{aligned}
J(\pi^*) - J(\widehat{\pi}^t) &\leq \eta \mathbb{E}^{\pi^t} \sum_{h=1}^{H} \left[ \left( \sum_{j=h}^{H} (\widehat{Q}_j^t(s_j, a_j) - \mathcal{T}_\eta^j \widehat{V}_{j+1}^t(s_j, a_j)) \right)^2 \right] \\
&\leq \eta H \mathbb{E}^{\pi^t} \sum_{h=1}^{H} \sum_{j=h}^{H} \left[ \left( \widehat{Q}_j^t(s_j, a_j) - \mathcal{T}_\eta^j \widehat{V}_{j+1}^t(s_j, a_j) \right)^2 \right] \\
&\leq \eta H^2 \mathbb{E}^{\pi^t} \sum_{h=1}^{H} \left[ \left( \widehat{Q}_h^t(s_h, a_h) - \mathcal{T}_\eta^h \widehat{V}_{h+1}^t(s_h, a_h) \right)^2 \right],
\end{aligned}$$

where the second inequality applies the Jensen's inequality.

Then, by Lemma B.2, with probability at least $1 - \delta$, we have for all $t \in [T]$ and $h \in [H]$,

$$\widehat{Q}_h^t(s_h, a_h) - \mathcal{T}_\eta^h \widehat{V}_{h+1}^t(s_h, a_h) = b_h^t(s_h, a_h) + \widehat{f}_h^t(s_h, a_h) - \mathcal{T}_\eta^h \widehat{V}_{h+1}^t(s_h, a_h)$$

$$\leq b_h^t(s_h, a_h) + \beta_h^T \cdot U_{\mathcal{F}_h}^h(s_h, a_h; \mathcal{D}_{t-1}^h)$$

$$= 2\beta_h^T \cdot U_{\mathcal{F}_h}^h(s_h, a_h; \mathcal{D}_{t-1}^h),$$

where the last equality uses the definition of bonus $b_h^t$.

Therefore, we have

$$\text{Regret}(T) = \sum_{t=1}^{T} \big[J(\pi^*) - J(\widehat{\pi}^t)\big]$$

$$\leq 4\eta H^2 \sum_{t=1}^{T} \sum_{h=1}^{H} (\beta_h^T)^2 \mathbb{E}_{s_1 \sim d_0, \widehat{\pi}}[(U_{\mathcal{F}_h}^h(s_h, a_h; \mathcal{D}_{t-1}^h))^2]$$

$$= \mathcal{O}\Big(\eta H^2 d(\mathcal{F}, \lambda, T) \cdot O(\log(N_{\mathcal{F}} T H / \delta))\Big),$$

where $d(\mathcal{F}, \lambda, T) = \sum_{h=1}^{H} d(\mathcal{F}_h, \lambda, T)$. $\qquad\square$

## C. Proof of Technical Lemmas

*Proof of Lemma A.2.* For the first equation, we notice that

$$\frac{\partial Z_R(x)}{\partial R(x, a)} = \frac{\partial \sum_{a' \in \mathcal{A}} \pi_{\text{ref}}(a'|x) \exp\big(\eta R(x, a')\big)}{\partial R(x, a)},$$

$$= \frac{\partial \pi_{\text{ref}}(a|x) \exp\big(\eta R(x, a)\big)}{\partial R(x, a)},$$

$$= \eta \pi_{\text{ref}}(a|x) \exp\big(\eta R(x, a)\big).$$

For the second equation, with the derivation rule of $\big(\frac{f(z)}{g(z)}\big)' = \frac{f'(z)g(z) - f(z)g'(z)}{(g(z))^2}$, we have

$$\frac{\partial \pi_R^\eta(a|x)}{\partial R(x, a)} = \frac{\partial \big[\pi_{\text{ref}}(a|x) \exp\big(\eta R(x, a)\big)/Z_R(x)\big]}{\partial R(x, a)}$$

$$= \frac{\eta \pi_{\text{ref}}(a|x) \exp\big(\eta R(x, a)\big) Z_R(x) - \partial Z_R(x) \cdot \pi_{\text{ref}}(a|x) \exp\big(\eta R(x, a)\big)}{Z_R(x)^2}$$

$$= \eta \pi_R^\eta(a|x) - \eta \pi_R^\eta(a|x)^2,$$

where we use the first equation in the last step. Finally, for the last equation, we have

$$\frac{\partial \pi_R^\eta(a'|x)}{\partial R(x, a)} = \frac{\partial \big[\pi_{\text{ref}}(a'|x) \exp\big(\eta R(x, a')\big)/Z_R(x)\big]}{\partial R(x, a)}$$

$$= -\frac{\pi_{\text{ref}}(a'|x) \exp\big(\eta R(x, a')\big) \cdot \partial Z_R(x)}{Z_R(x)^2}$$

$$= -\eta \pi_R^\eta(a|x) \cdot \pi_R^\eta(a'|x).$$

$\qquad\square$

We follow Theorem 13.15 from Zhang (2023) and Ye et al. (2023) to derive the in-sample error for the function.

**Lemma C.1.** *Consider a function space $\mathcal{F} : \mathcal{Z} \to \mathbb{R}$ and a filtered sequence $\{z_t, \epsilon_t\} \in \mathcal{Z} \times \mathbb{R}$ so that $\epsilon_t$ is conditional zero-mean $\eta$-sub-Gaussian noise. Suppose that $\mathcal{F}$ is a finite space with cardinality $N_{\mathcal{F}}$ For $f_*(\cdot) : \mathcal{Z} \to \mathbb{R}$, suppose that $y_t = f_*(x_t, a_t) + \epsilon_t$. If $\widehat{f}_t$ is an ERM solution:*

$$\widehat{f}_t = \underset{f \in \mathcal{F}}{\arg\min} \sum_{i=1}^{t} (f(z_i) - y_i)^2,$$

*with probability at least $1 - \delta$, we have for all $t \in [T]$,*

$$\sum_{i=1}^{t}(\widehat{f}_t(z_i) - f_*(z_i)^2 \leq 8\eta^2 \log(N_{\mathcal{F}}T/\delta).$$

*Proof.* For $f \in \mathcal{F}$, define

$$
\begin{aligned}
\phi(f, z_t) &= -a\left[(f(z_t) - y_t)^2 - (f_*(z_t) - y_t)^2\right] \\
&= -a\left[(f(z_t) - f_*(z_t) - \epsilon_t)^2 - \epsilon_t^2\right] \\
&= 2a(f(z_t) - f_*(z_t)) \cdot \epsilon_t - a(f(z_t) - f_*(z_t))^2,
\end{aligned}
$$

where $a = \eta^{-2}/4$. Since $\epsilon_t$ is $\eta$-sub-Gasussian conditional on $z_t, \mathcal{S}_{t-1} := \{z_i, \epsilon_i\}_{i \in [t-1]}$, we get

$$
\begin{aligned}
&\log \mathbb{E}_{y_t}\left[\exp\left(\phi(f, z_t) + a(f(z_t) - f_*(z_t))^2\right)\right] \\
&\leq \frac{4a^2(f(z_t) - f_*(z_t))^2\eta^2}{2} = \frac{(f(z_t) - f_*(z_t))^2}{8\eta^2},
\end{aligned}
$$

which means that

$$\log \mathbb{E}_{y_t}\left[\exp\left(\phi(f, z_t)\right)\right] \leq \frac{(f(z_t) - f_*(z_t))^2}{8\eta^2} - \frac{(f(z_t) - f_*(z_i))^2}{4\eta^2} = -\frac{(f(z_t) - f_*(z_t))^2}{8\eta^2}.$$

Then, by Lemma D.1 with $\lambda = 1$, we have for all $f \in \mathcal{F}$ and $t \in [T]$, with probability at least $1 - \delta$,

$$
\begin{aligned}
\sum_{i=1}^{t}\phi(f, z_i) &\leq \log \mathbb{E}_{y_t}\left[\exp\left(\phi(f, z_t)\right)\right] + \log(N_{\mathcal{F}}T/\delta) \\
&\leq -\frac{(f(z_t) - f_*(z_t))^2}{8\eta^2} + \log(N_{\mathcal{F}}T/\delta).
\end{aligned}
$$

Additionally, since $\widehat{f}_t$ is the ERM solution, we have

$$\sum_{i=1}^{t}\phi(\widehat{f}_t, z_i) = a\left[\sum_{i=1}^{t}(f_*(z_i) - y_i)^2 - \sum_{i=1}^{t}(\widehat{f}_t(z_i) - y_i)^2\right] \geq 0.$$

Combining the results above leads to

$$\sum_{i=1}^{t}(\widehat{f}_t(z_i) - f_*(z_i)^2 \leq 8\eta^2 \log(N_{\mathcal{F}}T/\delta).$$

$\square$

## D. Auxliary Lemmas

**Lemma D.1** ([Russo & Van Roy 2013](#)). *Consider a sequence of random variables $\{Z_t\}_{t \in \mathbb{N}}$ adapted to the filtration $\{\mathcal{S}_t\}_{t \in \mathbb{N}}$ and a function $f \in \mathcal{F}$. For any $\lambda > 0$, with probability at least $1 - \delta$, for all $t \geq 1$, we have*

$$-\sum_{s=1}^{t} f(Z_s) - \frac{1}{\lambda}\sum_{s=1}^{t}\log \mathbb{E}[e^{-\lambda f(Z_s)}|\mathcal{S}_{s-1}] \leq \frac{\log(1/\delta)}{\lambda}$$

*Proof.* For a detailed proof, see Theorem 13.2 of [Zhang](#) (2023) or Lemma 34 of [Foster & Rakhlin](#) (2023). $\square$

**Lemma D.2** (Online-to-batch conversion). *If an algorithm has a sublinear regret of $c^\dagger \cdot \log T$, then the algorithm finds an $\epsilon$-optimal policy with at most $\widetilde{\Theta}(c^\dagger/\epsilon)$ samples, where $\widetilde{\Theta}$ omit logarithmic terms of $c^\dagger/\epsilon$. Here $c^\dagger$ is a problem-dependent constant.*

*Proof.* We denote the policy sequence as $\{\pi^1, \cdots, \pi^T\}$. Then, by definition of regret, we know

$$\text{Regret}(T) = T \cdot V_1^*(x_1) - \sum_{t=1}^{T} V_1^{\pi^t}(x_1)$$

$$\leq c^\dagger \cdot \log T.$$

We consider the uniform policy $\widetilde{\pi} := \text{Uniform}(\pi^1, \cdots, \pi^T)$. It follows that

$$V_1^*(x_1) - V_1^{\widetilde{\pi}}(x_1) = V_1^*(x_1) - \frac{1}{T} \sum_{t=1}^{T} V_1^{\pi^t}(x_1) \leq c^\dagger \cdot \frac{\log T}{T}.$$

It suffices to prove that

$$c^\dagger \cdot \frac{\log T}{T} \leq \epsilon,$$

which is equivalent to solving

$$T \leq \exp(T \cdot \epsilon / c^\dagger).$$

By using the Lambert W function[2], we can prove that

$$T \geq \frac{W(1)c^\dagger}{\epsilon},$$

where $W(1) \geq \log(1/\epsilon) - \log\log(1/\epsilon)$.

$\square$

---

[2] https://en.wikipedia.org/wiki/Lambert_W_function

