# OpenReview forum: "Logarithmic Regret for Online KL-Regularized Reinforcement Learning"
_ICML.cc/2025/Conference — ICML 2025 poster_

### Official Review · Reviewer_cCx6 · 2025-03-11

**Overall Recommendation:** 4

**Summary:**

This paper studied online RL with KL regularization, and proposed an optimism-based algorithm for contextual bandits and RL. The paper showed that the regret scales logarithmic with the number of iterations, demonstrating the superiority of KL regularization used in RL. Particularly, the paper developed a new technique in the analysis of the regret, where it performs a refined analysis of the normalization term used in policy updating, and helps to show a reduced regret.

**Claims And Evidence:**

Yes. The paper provided detailed theoretical analysis to support their claims and theorems.

**Essential References Not Discussed:**

No.

**Experimental Designs Or Analyses:**

N/A

**Methods And Evaluation Criteria:**

Yes. This is a theoretical paper and theoretical analysis is provided.

**Other Comments Or Suggestions:**

In the claim of achieving logarithm regret (e.g. remark 4.2), it is better to explain it is because the optimal policy is the one that maximizes the KL-regularized objective, rather than standard objective (cumulative return). Since the lower bound of standard online RL is $\Omega(\sqrt{T})$. This could help to improve the clarity.

**Other Strengths And Weaknesses:**

Strength: the logarithm regret result seems promising and interesting. the technique for analyzing the normalization term is novel.

Weakness: Lack some (at least) simple experiments to demonstrate the proposed algorithm and validate the results.

**Questions For Authors:**

N/A

**Relation To Broader Scientific Literature:**

N/A

**Theoretical Claims:**

I briefly checked the proofs, particularly the analysis of the normalization term, and they seem correct to me.

---

> ### Author Rebuttal · Authors · 2025-04-01
>
> # Response to Reviewer cCx6
>
> Thank you for your strong support!
>
> **Q1** In the claim of achieving logarithm regret (e.g. remark 4.2), it is better to explain it is because the optimal policy is the one that maximizes the KL-regularized objective, rather than standard objective (cumulative return)
>
> **A1** Thanks for reminding this point! We will add this to the revision and state that now our objective is the reward subtracted by the KL-regualrization and is different from the original RL objective. We focus on this objective because as stated in Lines 21-34 right, for post-training in large-sacle models, the trained policy should not move too far from the strong base policy. Otherwise, the "sloignment tax" arises.
>
> ---
>
> **Q2** Lack some (at least) simple experiments to demonstrate the proposed algorithm and validate the results.
>
> **A2** Since large scale experiments are beyond the scope of this theoretical work, we run simulations for our algorithm under a multi-armed linear contextual bandit. We use different values of KL-regularization parameter $\eta$ and arm number $K$. The results are provided at this anonymous link https://anonymous.4open.science/r/Simulation_KL_RLHF-C092/KL_reg_MDPs.pdf. Since the x-axis has logarithmic scale, the almost linear curve in the figures validates that the regret scales logarithmically with rounds $T$.

---

### Official Review · Reviewer_yuSc · 2025-03-14

**Overall Recommendation:** 4

**Summary:**

Short summary: the authors propose a KL-regularized contextual bandit algorithm, and show it achieves logarithmic regret using a fine-grained analysis of the sub-optimality gap. They then extend this analysis to KL-regularized Reinforcement Learning by reducing to the bandit setting. The resulting KL-regularized least squares value iteration algorithm achieves a regret logarithmic in the number of rounds and polynomial in the horizon.

More detailed summary of bandit section:
- The proposed algorithm consists of using a least-squares fit of received rewards, and using as a policy a Gibbs reweighting of the reference policy according to the reward approximation plus an exploration bonus (i.e. $\pi(a|x) \propto \pi_{ref}(a|x) e^{\eta(\hat{R}(x,a) + b(x,a)}$).
- The exploration bonus is a monotonic function of reward uncertainty (Def. 3.3).
- Contrary to prior works, which apply a classical suboptimality gap decomposition in the KL-regularized setting, the authors propose a fine-grained analysis of the KL-regularized suboptimality gap. This analysis directly considers the normalization term for the policy ($Z_R(x) = \sum_a \pi_{ref}(a|x) e^{\eta R(x, a)}$).
- They can then bound the regret by the sum of squared bonuses, from which they claim their regret bound follows

More detailed summary of RL section:
- The proposed bandit algorithm is modified by replacing the least-squares fit of the reward by least-squares value iteration (i.e. learning a Q-function estimator $\hat{f}$ to minimize the square Bellman backward error $|| \hat{f}(s, a) - r_h - \hat{V}^{\hat{f}}(s)||^2$), still using an exploration bonus.
- To reduce to the bandit setting, the authors consider couplings of the learned policy $\hat{\pi}$ and the optimal policy $\pi^*$. A coupled policy $\hat{\pi}^{(h)}$ acts according to $\hat{\pi}$ until episode timestep $h$, and subsequently according to $\pi^*$.
- Since $\hat{\pi}^{(0)} = \pi^*$ and $\hat{\pi}^{(H)} = \hat{\pi}$ (where $H$ is the episode horizon), the suboptimality gap can be decomposed as a telescoping sum of $\mathbb{E}[V^{\hat{\pi}^{(h)}}(s) - V^{\hat{\pi}^{(h+1)}}(s)]$.
- Since each term in the telescoping sum depends only on one timestep of the environment, the analysis from the bandit case (i.e. $H=1$) applies. This yields a bound of the suboptimality gap by $H^2$ times the sum of squared approximate Bellman errors (i.e. according to the algorithm’s estimates $\hat{f}, \hat{Q}, \hat{V}$).
- Since the proposed algorithm minimizes squared approximate Bellman errors, standard function approximation generalization bounds imply the regret is logarithmic in the number of episodes.

**Claims And Evidence:**

The authors give a clear and comprehensible explanation of their proposed algorithms and proofs. I believe it would have been helpful if, when referencing “standard techniques” (e.g. lines 368-369), the authors would have provided specific citations or references to pin down which techniques exactly they have in mind. But, besides this, I found the core technical claims to be well explained and justified.

**Essential References Not Discussed:**

I am not aware of missing essential references, but will revisit this during the discussion period in case any are pointed out by other reviewers.

**Experimental Designs Or Analyses:**

N/A

**Methods And Evaluation Criteria:**

This is a theory paper proving a regret bound, and so experimental evaluations are not of major relevance in this case.

**Other Comments Or Suggestions:**

- There is a large whitespace in Line 231, second column. It might be best to not leave the corresponding equation as an inline expression.
- It would be helpful if the authors could also reference the rationale behind the inequality in Line 365.

**Other Strengths And Weaknesses:**

Other strengths:
- Clarity: the proof outline in the main paper clearly conveys the main techniques introduced, and is comprehensible to non-experts in relevant preceding works.

Other weaknesses (or, rather, interesting further directions I would have been interested in seeing in this paper, but can each perfectly well be left for future work):
- It would have been interesting to see computational resource considerations in the analysis of KL-LSVI. For example, in RLHF applications, reward models can be large-scale neural networks. In this case, optimizing a reward model at every episode could be very costly.
- In the same direction as the above, it would be interesting to see generalizations of KL-LSVI to the setting where $\hat{f}$ is a neural network, e.g. optimized via gradient descent during training. I would imagine Line 5 might be replaced e.g. by a  stochastic gradient update. I wonder if there are any standard techniques that could be used to generalize this paper’s analysis to such an online SGD setting.

**Questions For Authors:**

N/A

**Relation To Broader Scientific Literature:**

Given that the prior state-of-the-art regret bound for KL-regularized RL was $O(\sqrt{T})$, and that the authors prove a logarithmic bound, the result seems to be of major theoretical significance. In addition, as highlighted in Table 1, the author’s proposed algorithm achieves $O(1/\epsilon)$ sample complexity, while prior art achieves only $O(1/\epsilon^2)$.

Further, the “policy decomposition” technique used in line 422 to reduce the RL setting to the bandit setting seems fairly general, and might be applicable in other analyses of regret in MDPs.

**Theoretical Claims:**

I reviewed the elements of the proof presented in the main paper, but did not go into the ones in the supplementary materials in detail.

---

> ### Author Rebuttal · Authors · 2025-04-01
>
> # Response to Reviewer yuSc
>
> Thank you very much for your strong support!
>
> **Q1** It would have been interesting to see computational resource considerations in the analysis of KL-LSVI. For example, in RLHF applications, reward models can be large-scale neural networks. In this case, optimizing a reward model at every episode could be very costly.
>
> **A1** Our work focuses on providing a general and standard online framework enjoying logarithmic regret bounds. The computational efficiency is indeed very important for larges-scale models. Hence, to improve efficiency in RLHF applications, it has been shown that low switching techniques [1,2,3] can solve this problem both emprirically and theoretically. Following the batch framework in [1], we can obtain the same sample complexity with $\Theta(d_{\mathcal R})$ number of batches.
>
> ---
>
> **Q2** It would be interesting to see generalizations of KL-LSVI to the setting where
>  is a neural network, e.g. optimized via gradient descent during training. I would imagine Line 5 might be replaced e.g. by a stochastic gradient update. I wonder if there are any standard techniques that could be used to generalize this paper’s analysis to such an online SGD setting.
>
> **A2** In practice, a promising direction is Thompson Sampling (TS)-based exploration, which can be implemented using Stochastic Gradient Langevin Dynamics (SGLD) to introduce randomness into SGD for regression. SGLD enables implicit posterior sampling during training, potentially improving the balance between exploration and exploitation in the KL-regularized setting.
>
> To extend the current theoretical analysis of KL-LSVI, one could explore whether existing regret bounds for Langevin-based exploration methods apply in this context. Notably, [4] shows that TS with a "feel-good" term achieves an optimal sample complexity bound comparable to UCB in contextual bandits with function approximation. Furthermore, [5] provides a comprehensive empirical study demonstrating the efficiency of TS in RL and the effectiveness of SGLD in approximating TS. We believe investigating sampling-based methods for KL-regularized objectives in RL is an interesting direction for future work.
>
> ---
>
> **Q3** I believe it would have been helpful if, when referencing “standard techniques” (e.g. lines 368-369), the authors would have provided specific citations or references to pin down which techniques exactly they have in mind.
>
> **A3** Thanks for the helpful suggestion! We will explain the standard techniques detailedly in the revision.
>
>
> [1] Xiong W, et al. Iterative preference learning from human feedback: Bridging theory and practice for rlhf under kl-constraint. ICML, 2023.
>
> [2] Bai, Y., et al. Training a helpful and harmless assistant with reinforcement learning from human feedback.
>
> [3] Touvron, H., et al. Llama 2: Open foundation and fine-tuned chat models.
>
> [4] Zhang, T. Feel-good thompson sampling for contextual bandits and reinforcement learning.
>
> [5] Ishfaq, H., et al. Provable and Practical: Efficient Exploration in Reinforcement Learning via Langevin Monte Carlo

---

### Official Review · Reviewer_TRX2 · 2025-03-14

**Overall Recommendation:** 4

**Summary:**

This paper considers the problem of online KL-regularized contextual bandits and MDPs and proposes two provably efficient algorithms with logarithmic regret bounds, improving over the typical $O(\sqrt{T})$ regret bounds.

The key idea is a refined value/policy decomposition technique for the bandits/MDPs with KL-regularization.

## update after rebuttal

I maintain my positive score.

**Claims And Evidence:**

Yes

**Essential References Not Discussed:**

None that I know of.

**Experimental Designs Or Analyses:**

N/A

**Methods And Evaluation Criteria:**

N/A

**Other Comments Or Suggestions:**

N/A

**Other Strengths And Weaknesses:**

Strengths:
I believe this is a strong result and the ideas used here may also be of independent interest for other works.

Weakness:
There are no experiments to demonstrate the effectiveness of the proposed algorithms in practice.

**Questions For Authors:**

In page 4, it is stated that "Without loss of generality, we assume that the function class has finite cardinality" with a reference to a ~500 page book. How should the proof be modified to handle the infinite case?
In order to make the claim that there is no loss in generality, I suggest adding an appendix section to at least state the exact regret bound in the infinite case and include a brief overview of how it could be obtained, with more detailed references.

**Relation To Broader Scientific Literature:**

The related works section is comprehensive and covers the relation to the broader literature.

**Theoretical Claims:**

I didn't check all the details, but as far as I have checked, the proofs are correct.

---

> ### Author Rebuttal · Authors · 2025-04-01
>
> # Response to Reviewer TRX2
>
> Thank you for your strong support!
>
> **Q1** In page 4, it is stated that "Without loss of generality, we assume that the function class has finite cardinality" with a reference to a ~500 page book. How should the proof be modified to handle the infinite case? In order to make the claim that there is no loss in generality, I suggest adding an appendix section to at least state the exact regret bound in the infinite case and include a brief overview of how it could be obtained, with more detailed references.
>
> **A1** Thank you for your helpful suggestion! For the infinite case, we will assume that the function class has finite covering number. For each function $f\in\mathcal R$, there exists a cover function $f'$ belongs to the cover such that $f$ and $f'$ are close enough so that we can transform this problem into a finite class and take the uniform bound. Then, in the final regret bound, we just need to replace the class cardinality with the covering number. This analysis is standard in previous literature [1,2] and Chapter 4.6 of Zhang, 2023. We will provide the details in the revision.
>
>
> [1] Zhao H, et al. Sharp analysis for kl-regularized contextual bandits and rlhf.
>
> [2] Ye C, et al. Corruption-robust algorithms with uncertainty weighting for nonlinear contextual bandits and markov decision processes. ICML, 2023.

---

### Official Review · Reviewer_SUKH · 2025-03-15

**Overall Recommendation:** 3

**Summary:**

The authors noted that the theoretical differences between KL-regularized reinforcement learning (RL) and standard RL have not been thoroughly explored. Recent studies analyzing KL-regularized objectives in decision-making either revert to traditional RL settings or depend on strong coverage assumptions. In the KL-regularized contextual bandits and Markov Decision Processes (MDPs) within the online RL framework, the authors proposed KL-regularized UCB and KL-regularized LSVI with UCB. These two algorithms are based on the standard optimism principle and demonstrate that they achieve regret bounds that scale logarithmically with the number of rounds, T.

## update after rebuttal
Thanks to the authors' efforts in providing feedback. For the simulated experiments conducted by the authors, it is recommended that they include detailed information about the experimental settings related to the controllable synthetic environment. This information should be provided in the supplementary material for clarity and transparency.

**Claims And Evidence:**

The proposed algorithms, KL-regularized UCB and KL-regularized LSVI with UCB, are based on theoretical analyses and have yet to undergo empirical validation.

**Essential References Not Discussed:**

None

**Experimental Designs Or Analyses:**

No related experimental designs or analyses.

**Methods And Evaluation Criteria:**

This work did not include any practical experiments for the evaluation of the algorithms.

**Other Comments Or Suggestions:**

None

**Other Strengths And Weaknesses:**

None

**Questions For Authors:**

1. The two proposed algorithms are said to achieve regret bounds that scale logarithmically with the number of rounds, T. To support these claims, empirical validations should be conducted in real-world Reinforcement Learning from Human Feedback (RLHF) experiments to provide valuable insights. It is curious whether any empirical validations related could be provided.
2. Both algorithms include bonus terms, specifically a constant value of 1 found in equations (3) and (9). How does the constant affect the uncertainty associated with this bonus and the reward function? Additionally, is it possible to explore other potential bonus terms that could be utilized within these two algorithms?

**Relation To Broader Scientific Literature:**

The authors provide two provably efficient algorithms: KL-regularized UCB and KL-regularized LSVI with UCB. Both algorithms theoretically achieve the logarithmic regret bound.

**Theoretical Claims:**

The theoretical analysis outlined in the main paper seems to be well-structured and thorough at first glance.

---

> ### Author Rebuttal · Authors · 2025-04-01
>
> # Response to Reviewer SUKH
>
> Thank you for your insightful comments! We address your questions as follows.
>
> **Q1** The two proposed algorithms are said to achieve regret bounds that scale logarithmically with the number of rounds, T. To support these claims, empirical validations should be conducted in real-world Reinforcement Learning from Human Feedback (RLHF) experiments to provide valuable insights. It is curious whether any empirical validations related could be provided.
>
> **A1**: Thanks for the point. We would like to clarify that the main focus of this project is to understand the statistical limit of the online decision making under the KL-regularized objective. Past experience suggests these understandings can be effectively connected to the downstream real-world algorithmic designs in principle, though with certain heuristic approximation to overcome the gap between the theoretical side and the empirical side. Even in the RLHF literature, we notice that the theoretical advantage of iterative RLHF/DPO was first established in [1] and then its empirical recipe is presented in [2, 3] and became standard practice in the past year. The idea of pessimism was established in [4], and empirical community construct reward model with uncertainty penalization to mitigate the reward hacking issue [5].
>
> We hope that the insights from this work can motivate a comprehensive study with real-world experiments for future work, which is beyond the scope of this work. Meanwhile, a direct validation of the faster rate with real-world experiments is also not feasible since we have no access to the groud-truth reward.
>
> However, we agree that empirical study can help to make the study complete. Therefore, we conduct some simulated experiments under controllable synthetic environment. Specifically, we run simulations for our algorithm under a multi-armed linear contextual bandit and different values of KL-regularization parameter $\eta$. The result provided at this anonymous link https://anonymous.4open.science/r/Simulation_KL_RLHF-C092/KL_reg_MDPs.pdf validates that the regret scale logarithmically with rounds $T$.
>
> ---
>
> **Q2** Both algorithms include bonus terms, specifically a constant value of 1 found in equations (3) and (9). How does the constant affect the uncertainty associated with this bonus and the reward function? Additionally, is it possible to explore other potential bonus terms that could be utilized within these two algorithms?
>
> **A2**: Thanks for the questions. The central idea here is that we should use an optimistically biased objective to facilitate exploration of the underlying space, where adding an uncertainty bonus is one of the most common approach in the literature. The upper bound of the bonus may be very large compared to the range of the reward, so we usually truncate the bonus by $1$.
>
> From the theoretical perspective, which is also the main focus of our paper, such as bonus should ensure the estimator is an upper bound of the true parameter with high probability and is constructed mostly by the concentration inequality tool from the high-dimensional statistics, where the constants are determined accordingly to satisfy this goal. Meanwhile, any potential bonus that satisfies this condition is valid but we usually take the sharpest estimation for a better theoretical results.
>
> When we move to the practical side, we typically resort to heuristic approximation of the bonus and apply the theoretical insights in principle. For instance, a common way is to use the ensemble method to incorporate the uncertainty into empirical experiments [5]. There are also alternative ways that can be used as "potential bonus" like using a optimistically biased loss function [6, 7]. Nevertheless, the empirical results in these works consistently show that taking uncertainty into consideration can largely improve the real-world performance of the algorithms.
>
> [1] Xiong, W, et al. Iterative preference learning from human feedback: Bridging theory and practice for rlhf under kl-constraint. ICML, 2024.
>
> [2] Guo, S, et al. Direct Language Model Alignment from Online AI Feedback. 2024.
>
> [3] Dong, H, et al. RLHF Workflow: From Reward Modeling to Online RLHF. TMLR, 2024.
>
> [4] Jin, Y, et al. Is pessimism provably efficient for offline rl? ICML, 2021.
>
> [5] Coste, T, et al. Reward model ensembles help mitigate overoptimization. ICLR, 2024.
>
> [6] Xie, T, et al. Exploratory Preference Optimization: Harnessing Implicit Q*-Approximation for Sample-Efficient RLHF. 2024.
>
> [7] Liu, Z, et al. Provably mitigating overoptimization in rlhf: Your sft loss is implicitly an adversarial regularizer. NIPS, 2024.

---

### Official Review · Reviewer_b9cv · 2025-03-24

**Overall Recommendation:** 4

**Summary:**

The paper presents new high-probability regret bounds for entropy-regularized contextual bandits and finite-horizon reinforcement learning. Concretely, the authors show that the regret bound is logarithmic in the time horizon, which improves asymptotically on existing bounds.

**Claims And Evidence:**

The main claim is in the form of high-probability regret bounds, and the evidence is in the form of a theoretical analysis that supports the claim.

**Essential References Not Discussed:**

I am not aware of any.

**Experimental Designs Or Analyses:**

N/A

**Methods And Evaluation Criteria:**

Entropy-regularization is often used in reinforcement learning applications, but this paper is purely theoretical.

**Other Comments Or Suggestions:**

The name KL-UCB already exists in the literature, and I would strongly advice the authors to change the name of their novel algorithms.

I think the introduction makes excessive references to RLHF. Though LLMs have provided a successful application domain for reinforcement learning in recent years, entropy regularization is commonly used with reinforcement learning in a wide range of domains, and there is nothing specific to RLHF in the proposed algorithms.

In the first paragraph of related work, there are two almost identical sentences about PPO for LLMs.

**Other Strengths And Weaknesses:**

In terms of strengths, the analysis depends on novel regret decompositions which may be of interest in other areas. In particular, I had never seen the regret decomposition that changes the policy for one time step at a time, which allows bounding the local regret as the expected square difference in value functions at that time step.

In terms of weaknesses, I miss a discussion of the eluder dimension as mentioned above.

**Questions For Authors:**

What is the exact dependence on the horizon H?

Are there known results regarding the eluder dimension for the class of neural networks used to train LLMs?

**Relation To Broader Scientific Literature:**

The theoretical results of the paper shed light on why entropy regularization is commonly used in successful applications of reinforcement learning.

**Theoretical Claims:**

I did check the proof outline, and to the best of my knowledge the theoretical claims are correct, though I did not check the proof of each individual lemma in detail.

After defining the eluder dimension and providing an example, the authors do not discuss it further. However, the overall regret bound is only logarithmic in T if the eluder dimension is logarithmic in T. I understand that there are function classes for which this is true, but the authors need to discuss the implications of this, especially since they promote entropy-regularization for large-scale RLHF.

In finite-horizon reinforcement learning it is usually assumed that the value function is bounded by H rather than 1. The authors do not make it clear what the exact dependence on H is if we change this assumption (H^2?)

---

> ### Author Rebuttal · Authors · 2025-04-01
>
> # Response to Reviewer b9cv
>
> Thank you for your positive feedback! We answer your questions point-by-point.
>
> **Q1** After defining the eluder dimension and providing an example, the authors do not discuss it further. However, the overall regret bound is only logarithmic in $T$ if the eluder dimension is logarithmic in $T$. I understand that there are function classes for which this is true, but the authors need to discuss the implications of this, especially since they promote entropy-regularization for large-scale RLHF.
>
> **A1** Thank you for the insightful comment! Eluder dimension plays an important role in our analysis for the regret bound. Here we provide a discussion of the eluder dimension as follows. The concept of eluder dimension is first introduced in [1], which serves as an important complexity measure for studying reinforcement learning with general function classes. It has been shown that for a (generalized) linear function class, the eluder dimension is $\sim d\log T$ where $d$ is the dimension of the function class [2]. Our logarithmic-in-$T$ regret bound holds when the eluder dimension grows at most logarithmically with $T$. We will add the detailed discussion and the clarification in our revision.
>
> ---
>
> **Q2** In finite-horizon reinforcement learning it is usually assumed that the value function is bounded by $H$ rather than 1. The authors do not make it clear what the exact dependence on $H$ is if we change this assumption?
>
> **A2** Thank you for your insightful question! If the value function is assumed to be bounded by $H$, then $e_j$ in Lemma 5.2 (on page 8) will be $O(H)$ times larger, which further yields an $O(\eta H^4\cdot d \log \mathcal{N})$ regret bound after applying Lemma 5.2. The additional dependence can be removed via variance weighting and the Bernstein inequality, which is, however, not the focus of our work. We will add this disscussion in the revision.
>
>
> [1] Russo, D. and Van Roy, B. Eluder dimension and the sample complexity of optimistic exploration. NeurIPS 2013
>
> [2] Wang, Ruosong, Russ R. Salakhutdinov, and Lin Yang. Reinforcement learning with general value function approximation: Provably efficient approach via bounded eluder dimension. NeurIPS 2020.

---

### Decision · Program_Chairs · 2025-05-01

**Decision:**

Accept (poster)

**Comment:**

This paper considers online learning using a KL-regularized objective in the settings of contextual bandits and reinforcement learning. It establishes, for the first time, regret bounds that scale logarithmically with the number of time steps $T$. Previous bounds scaled with $\sqrt{T}$ and relied on strong coverage assumptions.

All five reviewers agreed that this is a good paper with novel contributions and recommended acceptance. While some expressed concerns about the lack of experimental evaluation, I believe the theoretical contribution is strong and significant in its own right. Moreover, the decomposition introduced in one of the sections may be of independent interest. Based on this, I recommend acceptance.

The authors should address the following points in the revised version, in order of priority:

- To avoid confusion, consider resolving the algorithm naming conflict: the same name KL-UCB was already used in [1].
- Explicitly clarify the conditions required on the growth of the eluder dimension for the regret bounds to hold.
- Include the brief discussion on how to remove the additional dependence on the horizon $H$, which is currently left for future work but outlined in the rebuttal.
- Provide detailed information about the simple experiment that was shared during the author-reviewer discussion.

[1] *The KL-UCB Algorithm for Bounded Stochastic Bandits and Beyond*, A. Garivier and O. Cappé, ALT 2011.